# Guide Your Agent with Adaptive Multimodal Rewards

**Changyeon Kim**[1]    **Younggyo Seo**[2]    **Hao Liu**[3]    **Lisa Lee**[4]
**Jinwoo Shin**[1]    **Honglak Lee**[5,6]    **Kimin Lee**[1]

[1]KAIST    [2]Dyson Robot Learning Lab    [3]UC Berkeley
[4]Google DeepMind    [5]University of Michigan    [6]LG AI Research

## Abstract

Developing an agent capable of adapting to unseen environments remains a difficult challenge in imitation learning. This work presents Adaptive Return-conditioned Policy (ARP), an efficient framework designed to enhance the agent's generalization ability using natural language task descriptions and pre-trained multimodal encoders. Our key idea is to calculate a similarity between visual observations and natural language instructions in the pre-trained multimodal embedding space (such as CLIP) and use it as a reward signal. We then train a return-conditioned policy using expert demonstrations labeled with multimodal rewards. Because the multimodal rewards provide adaptive signals at each timestep, our ARP effectively mitigates the goal misgeneralization. This results in superior generalization performances even when faced with unseen text instructions, compared to existing text-conditioned policies. To improve the quality of rewards, we also introduce a fine-tuning method for pre-trained multimodal encoders, further enhancing the performance. Video demonstrations and source code are available on the project website: https://sites.google.com/view/2023arp.

## 1 Introduction

Imitation learning (IL) has achieved promising results in learning behaviors directly from expert demonstrations, reducing the necessity for costly and potentially dangerous interactions with environments [32, 59]. These approaches have recently been applied to learn control policies directly from pixel observations [7, 36, 57]. However, IL policies frequently struggle to generalize to new environments, often resulting in a lack of meaningful behavior [14, 58, 68, 79] due to overfitting to various aspects of training data. Several approaches have been proposed to train IL agents capable of adapting to unseen environments and tasks. These approaches include conditioning on a single expert demonstration [18, 20], utilizing a video of human demonstration [6, 76], and incorporating the goal image [16, 23]. However, such prior methods assume that information about target behaviors in test environments is available to the agent, which is impractical in many real-world problems.

One alternative approach for improving generalization performance is to guide the agent with natural language: training agents conditioned on language instructions [7, 50, 72]. Recent studies have indeed demonstrated that text-conditioned policies incorporated with large pre-trained multimodal models [22, 56] exhibit strong generalization abilities [46, 66]. However, simply relying on text representations may fail to provide helpful information to agents in challenging scenarios. For example, consider a text-conditioned policy (see Figure 1a) trained to collect a coin, which is positioned at the end of the map, following the text instruction "collect a coin". When we deploy the learned agent to test environments where the coin's location is randomized, it often fails to collect the coin. This is because, when relying solely on expert demonstrations, the agent might mistakenly think that the goal is to navigate to the end of the level (see supporting results in Section 4.1). This example shows the simple text-conditioned policy fails to fully exploit the provided text instruction and suffers from goal misgeneralization (i.e., pursuing undesired goals, even when trained with a correct specification) [15, 64].

37th Conference on Neural Information Processing Systems (NeurIPS 2023).

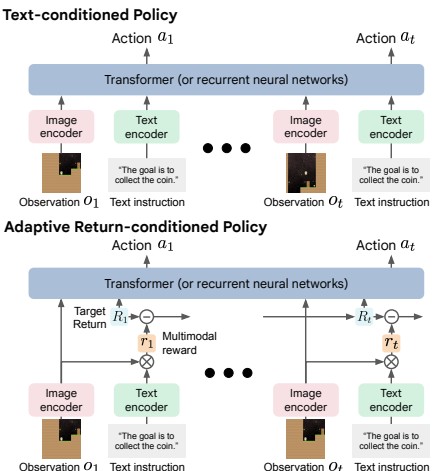

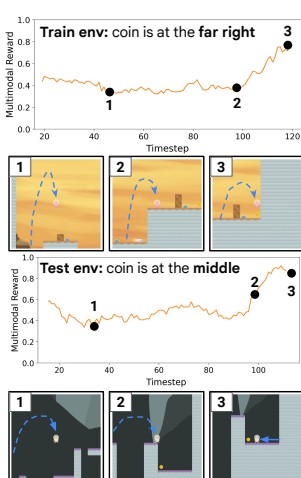

(a) Comparison of ARP with baseline    (b) Multimodal reward curve from CoinRun

Figure 1: (a) ARP utilizes the similarity between visual observations and text instructions in the pre-trained multimodal representation space as a reward and then trains the return-conditioned policy using demonstrations with multimodal reward labels. (b) Curves of multimodal reward for fine-tuned CLIP [56] in the trajectory from CoinRun environment. Multimodal reward consistently increases as the agent approaches the goal, and this trend remains consistent regardless of the training and test environment, suggesting the potential to guide agents toward target objects in test environments.

In this paper, we introduce Adaptive Return-conditioned Policy (ARP), a novel IL method designed to enhance generalization capabilities. Our main idea is to measure the similarity between visual observations and natural language task descriptions in the pre-trained multimodal embedding space (such as CLIP [56]) and use it as a reward signal. Subsequently, we train a return-conditioned policy using demonstrations annotated with these multimodal reward labels. Unlike prior IL work that relies on static text representations [46, 51, 67], our trained policies make decisions based on multimodal reward signals computed at each timestep (see the bottom figure of Figure 1a).

We find that our multimodal reward can provide a consistent signal to the agent in both training and test environments (see Figure 1b). This consistency helps prevent agents from pursuing unintended goals (i.e., mitigating goal misgeneralization) and thus improves generalization performance when compared to text-conditioned policies. Furthermore, we introduce a fine-tuning scheme that adapts pre-trained multimodal encoders using in-domain data (i.e., expert demonstrations) to enhance the quality of the reward signal. We demonstrate that when using rewards derived from fine-tuned encoders, the agent exhibits superior generalization performance compared to the agent with frozen encoders in test environments. Notably, we also observe that ARP effectively guides agents in test environments with unseen text instructions associated with new objects of unseen colors and shapes (see supporting results in Section 4.3).

In summary, our key contributions are as follows:

- We propose Adaptive Return-conditioned Policy (ARP), a novel IL framework that trains a return-conditioned policy using adaptive multimodal rewards from pre-trained encoders.

- We introduce a fine-tuning scheme that adapts pre-trained CLIP models using in-domain expert demonstrations to improve the quality of multimodal rewards.

- We show that our framework effectively mitigates goal misgeneralization, resulting in better generalization when compared to text-conditioned baselines. We further show that ARP can execute unseen text instructions associated with new objects of unseen colors and shapes.

- We demonstrate that our method exhibits comparable generalization performance to baselines that consume goal images from test environments, even though our method solely relies on natural language instruction.

- Source code and expert demonstrations used for our experiments are available at https://github.com/csmile-1006/ARP.git

## 2 Related Work

**Generalization in imitation learning**   Addressing the challenge of generalization in imitation learning is crucial for deploying trained agents in real-world scenarios. Previous approaches have shown improvements in generalization to test environments by conditioning agents on a robot demonstration [18, 20], a video of a human performing the desired task [76, 6], or a goal image [16, 44, 23]. However, these approaches have a disadvantage: they can be impractical to adopt in real-world scenarios where the information about target behaviors in test environments is not guaranteed. In this work, we propose an efficient yet effective method for achieving generalization even in the absence of specific information about test environments. We accomplish this by leveraging multimodal reward computed with current visual observations and task instructions in the pre-trained multimodal embedding space.

**Pre-trained representation for reinforcement learning and imitation learning**   Recently, there has been growing interest in leveraging pre-trained representations for robot learning algorithms that benefit from large-scale data [54, 73, 61, 52]. In particular, language-conditioned agents have seen significant advancements by leveraging pre-trained vision-language models [46, 66, 78, 38], drawing inspiration from the effectiveness of multimodal representation learning techniques like CLIP [56]. For example, *Instruct*RL [46] utilizes a pre-trained multimodal encoder [22] to encode the alignment between multiple camera observations and text instructions and trains a transformer-based behavior cloning policy using encoded representations. In contrast, our work utilizes the similarity between visual observations and text instructions in the pre-trained multimodal embedding space in the form of a reward to guide agents in the test environment adaptively.

We provide more discussions on related work in more detail in Appendix B.

## 3 Method

In this section, we introduce Adaptive Return-conditioned Policy (ARP), a novel IL framework for enhancing generalization ability using multimodal rewards from pre-trained encoders. We first describe the problem setup in Section 3.1. Section 3.2 introduces how we define the multimodal reward in the pre-trained CLIP embedding spaces and use it for training return-conditioned policies. Additionally, we propose a new fine-tuning scheme that adapts pre-trained multimodal encoders with in-domain data to enhance the quality of rewards in Section 3.3.

### 3.1 Preliminaries

We consider the visual imitation learning (IL) framework, where an agent learns to solve a target task from expert demonstrations containing visual observations. We assume access to a dataset $\mathcal{D} = \{\tau_i\}_{i=1}^{N}$ consisting of $N$ expert trajectories $\tau = (o_0, a_0^*, ..., o_T, a_T^*)$ where $o$ represents the visual observation, $a$ means the action, and $T$ denotes the maximum timestep. These expert demonstrations are utilized to train the policy via behavior cloning. As a single visual observation is not sufficient for fully describing the underlying state of the task, we approximate the current state by stacking consecutive past observations following common practice [53, 74].

We also assume that a text instruction $\mathbf{x} \in \mathcal{X}$ that describes how to achieve the goal for solving tasks is given in addition to expert demonstrations. The standard approach to utilize this text instruction is to train a text-conditioned policy $\pi(a_t | o_{\leq t}, \mathbf{x})$. It has been observed that utilizing pre-trained multimodal encoders (like CLIP [56] and M3AE [22]) is very effective in modeling this text-conditioned policy [46, 49, 66, 67]. However, as shown in the upper figure of Figure 1a, these approaches provide the same text representations regardless of changes in visual observations. Consequently, they would not provide the agent with adaptive signals when encountering previously unseen environments. To address this limitation, we propose an alternative framework that leverages $\mathbf{x}$ to compute similarity with the current visual observation within the pre-trained multimodal embedding space. We then employ this similarity as a reward signal. This approach allows the reward value to be adjusted as the visual observation changes, providing the agent with an adaptive signal (see Figure 1b).

### 3.2 Adaptive Return-Conditioned Policy

**Multimodal reward**   To provide more detailed task information to the agent that adapts over timesteps, we propose to use the visual-text alignment score from pre-trained multimodal encoders.

Specifically, we compute the alignment score between visual observation at current timestep $t$ and text instruction $\mathbf{x}$ in the pre-trained multimodal embedding space as follows:

$$r_{\phi,\psi}(o_t, \mathbf{x}) = s(f_\phi^{\text{vis}}(o_t), f_\psi^{\text{txt}}(\mathbf{x})).\tag{1}$$

Here, $s$ represents a similarity metric in the representation space of pre-trained encoders: a visual encoder $f_\phi^{\text{vis}}$ parameterized by $\phi$ and a text encoder $f_\psi^{\text{txt}}$ parameterized by $\psi$. While our method is compatible with any multimodal encoders and metric, we adopt the cosine similarity between CLIP [56] text and visual embeddings in this work. We label each expert state-action trajectory as $\tau^* = (R_0, o_0, a_0^*, ..., R_T, o_T, a_T^*)$ where $R_t = \sum_{i=t}^{T} r_{\phi,\psi}(o_i, \mathbf{x})$ denotes the multimodal return for the rest of the trajectory at timestep $t$[1]. The set of return-labeled demonstrations is denoted as $\mathcal{D}^* = \{\tau_i^*\}_{i=1}^N$.

**Return-conditioned policy**  Using return-labeled demonstrations $\mathcal{D}^*$, we train return-conditioned policy $\pi_\theta(a_t | o_{\leq t}, R_t)$ parameterized by $\theta$ using the dataset $\mathcal{D}^*$ and minimize the following objective:

$$\mathcal{L}_\pi(\theta) = \mathbb{E}_{\tau^* \sim \mathcal{D}^*}\left[ \sum_{t \leq T} l(\pi_\theta(a_t | o_{\leq t}, R_t), a_t^*) \right]\tag{2}$$

Here, $l$ represents the loss function, which is either the cross entropy loss when the action space is defined in discrete space or the mean squared error when defined in continuous space.

The main advantage of our method lies in its adaptability in the deployment by adjusting to multimodal rewards computed in test environments (see Figure 1a). At the test time, our trained policy predicts the action $a_t$ based on the target multimodal return $R_t$ and the observation $o_t$. Since the target return $R_t$ is recursively updated based on the multimodal reward $r_t$, it can provide a timestep-wise signal to the agent, enabling it to adapt its behavior accordingly. We find that this signal effectively guides the agent to prevent pursuing undesired goals (see Section 4.1 and Section 4.2), and it also enhances generalization performance in environments with unseen text instructions associated with objects having previously unseen configurations (as discussed in Section 4.3).

In our experiments, we implement two different types of ARP using Decision Transformer (DT) [8], referred to as ARP-DT, and using Recurrent State Space Model (RSSM) [25], referred to as ARP-RSSM. Further details of the proposed architectures are provided in Appendix A.

### 3.3 Fine-Tuning Pre-trained Multimodal Encoders

Despite the effectiveness of our method with pre-trained CLIP multimodal representations, there may be a domain gap between the images used for pre-training and the visual observations available from the environment. This domain gap can sometimes lead to the generation of unreliable, misleading reward signals. To address this issue, we propose fine-tuning schemes for pre-trained multimodal encoders $(f_\phi^{\text{vis}}, f_\psi^{\text{txt}})$ using in-domain dataset (expert demonstrations) $\mathcal{D}$ in order to improve the quality of multimodal rewards. Specifically, we propose fine-tuning objectives based on the following two desiderata: reward should (i) remain consistent within similar timesteps and (ii) be robust to visual distractions.

**Temporal smoothness**  To encourage the consistency of the multimodal reward over timesteps, we adopt the objective of value implicit pre-training (VIP) [52] that aims to learn smooth reward functions from action-free videos. The main idea of VIP is to (i) capture long-range dependency by attracting the representations of the first and goal frames and (ii) inject local smoothness by encouraging the distance between intermediate frames to represent progress toward the goal. We extend this idea to our multimodal setup by replacing the goal frame with the text instruction $\mathbf{x}$ describing the task objective and using our multimodal reward $R$ as below:

$$\mathcal{L}_{\text{VIP}}(\phi, \psi) = \underbrace{(1 - \gamma) \cdot \mathbb{E}_{o_1 \sim \mathcal{O}_1}[r_{\phi,\psi}(o_1, \mathbf{x})]}_{\text{long-range dependency loss}}$$
$$+ \underbrace{\log \mathbb{E}_{(o_t, o_{t+1}) \sim \mathcal{D}}[\exp(r_{\phi,\psi}(o_t, \mathbf{x}) + 1 - \gamma \cdot r_{\phi,\psi}(o_{t+1}, \mathbf{x}))]}_{\text{local smoothness loss}}.\tag{3}$$

---

[1]We assume the discount factor $\gamma$ as 1 in our experiments, and our method can also be applied in the setup with the discount factor less than 1.

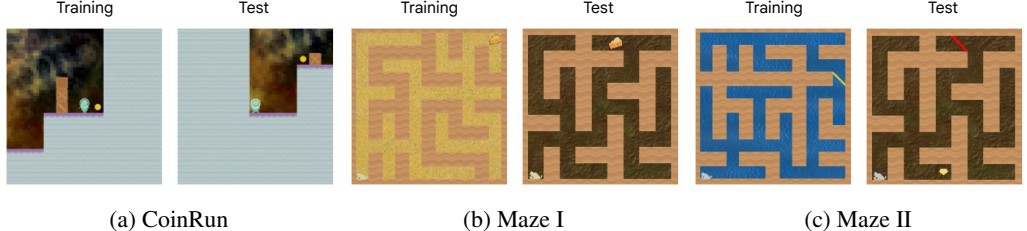

| (a) CoinRun | (b) Maze I | (c) Maze II |

Figure 2: Environments from OpenAI Procgen benchmarks [10] used in our experiments. We train our agents using expert demonstrations collected in environments with multiple visual variations. We then perform evaluations on environments from unseen levels with target objects in unseen locations. See Section 4.1 for more details.

Here $\mathcal{O}_1$ denotes a set of initial visual observations in $\mathcal{D}$. One can see that the local smoothness loss is the one-step temporal difference loss, which recursively trains the $r_{\phi,\psi}(o_t, \mathbf{x})$ to regress $-1 + \gamma \cdot r_{\phi,\psi}(o_{t+1}, \mathbf{x})$. This then induces the reward to represent the remaining steps to achieve the text-specified goal $\mathbf{x}$ [31], making rewards from consecutive observations smooth.

**Robustness to visual distractions** To further encourage our multimodal reward to be robust to visual distractions that should not affect the agent (*e.g.,* changing textures or backgrounds), we introduce the inverse dynamics model (IDM) objective [55, 33, 41]:

$$\mathcal{L}_{\text{IDM}}(\phi, \psi) = \mathbb{E}_{(o_t, o_{t+1}, a_t) \sim \mathcal{D}}[l(g(f_\phi^{\text{vis}}(o_t), f_\phi^{\text{vis}}(o_{t+1}), f_\psi^{\text{txt}}(\mathbf{x})), a_t^*)], \tag{4}$$

where $g(\cdot)$ denotes the prediction layer which outputs $\hat{a}_t$, predicted estimate of $a_t$, and $l$ represents the loss function which is either the cross entropy loss when the action space is defined in discrete space or the mean squared error when it's defined in continuous space. By learning to predict actions taken by the agent using the observations from consecutive timesteps, fine-tuned encoders learn to ignore aspects within the observations that should not affect the agent.

**Fine-tuning objective** We combine both VIP loss and IDM loss as the training objective to fine-tune pre-trained multimodal encoders in our model:

$$\mathcal{L}_{\text{FT}}(\phi, \psi) = \mathcal{L}_{\text{VIP}}(\phi, \psi) + \beta \cdot \mathcal{L}_{\text{IDM}}(\phi, \psi),$$

where $\beta$ is a scale hyperparameter. We find that both objectives synergistically contribute to improving the performance (see Table 7 for supporting experiments).

## 4 Experiments

We design our experiments to investigate the following questions:

1. Can our method prevent agents from pursuing undesired goals in test environments? (see Section 4.1 and Section 4.2)

2. Can ARP follow unseen text instructions? (see Section 4.3)

3. Is ARP comparable to goal image-conditioned policy? (see Section 4.4)

4. Can ARP induce well-aligned representation in test environments? (see Section 4.5)

5. What is the effect of each component in our framework? (see Section 4.6)

### 4.1 Procgen Experiments

**Environments** We evaluate our method on three different environments proposed in Di Langosco et al. [15], which are variants derived from OpenAI Procgen benchmarks [10]. We assess the generalization ability of trained agents when faced with test environments that cannot be solved without following true task success conditions.

- CoinRun: The training dataset consists of expert demonstrations where the agent collects a coin that is consistently positioned on the far right of the map, and the text instruction is "The goal

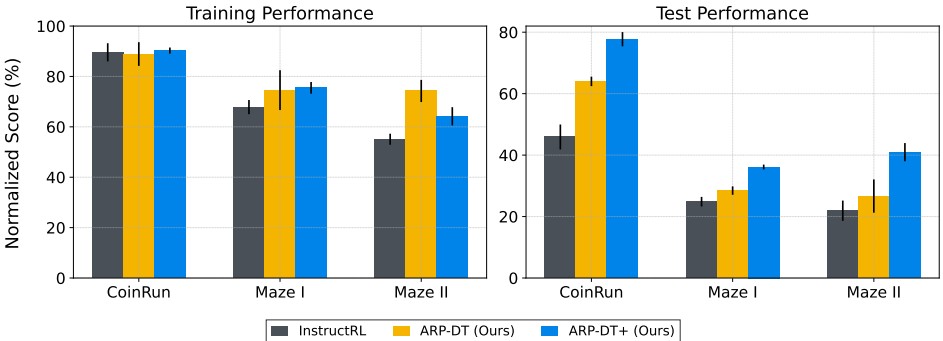

Figure 3: Expert-normalized scores on training/test environments. The result shows the mean and standard variation averaged over three runs. ARP-DT denotes the model that uses pre-trained CLIP representations, and ARP-DT+ denotes the model that uses fine-tuned CLIP representations (see Section 3.3) for computing the multimodal reward.

is to collect the coin.". Note the agent may mistakenly interpret that the goal is to proceed to the end of the level, as this also leads to reaching the coin when relying solely on the expert demonstrations. We evaluate the agent in environments where the coin's location is randomized (see Figure 2a) to verify that the trained agent truly follows the intended objective.

- Maze I: The training dataset consists of expert demonstrations where the agent reaches a yellow cheese that is always located at the top right corner, and the text instruction is "Navigate a maze to collect the yellow cheese.". The agent may misinterpret that the goal is to proceed to the far right corner, as it also results in reaching the yellow cheese when relying only on expert demonstrations. To verify that the trained agent follows the intended objective, we assess the trained agents in the test environment where the cheese is placed at a random position (see Figure 2b).

- Maze II: The training dataset consists of expert demonstrations where the agent approaches a yellow diagonal line located at a random position, and the text instruction is "Navigate a maze to collect the line.". The agent might misinterpret the goal as reaching an object with a yellow color because it also leads to collecting the object with a line shape when relying only on expert demonstrations, For evaluation, we consider a modified environment with two objects: a yellow gem and a red diagonal line. The goal of the agent is to reach the diagonal line, regardless of its color, to verify that the agent truly follows the intended objective (see Figure 2c).

**Implementation**  For all experiments, we utilize the open-sourced pre-trained CLIP model[2] with ViT-B/16 architecture to generate multimodal rewards. Our return-conditioned policy is implemented based on the official implementation of *Instruct*RL [46], and implementation details are the same unless otherwise specified. To collect expert demonstrations used for training data, we first train PPG [11] agents on 500 training levels that exhibit ample visual variations for 200M timesteps per task. We then gather 500 rollouts for CoinRun and 1000 rollouts for Maze in training environments. All models are trained for 50 epochs on two GPUs with a batch size 64 and a context length of 4. Our code and datasets are available at `https://github.com/csmile-1006/ARP.git`. Further training details, including hyperparameter settings, can be found in Appendix C.

**Evaluation**  We evaluate the zero-shot performance of trained agents in test environments from different levels (i.e., different map layouts and backgrounds) where the target object is either placed in unseen locations or with unseen shapes. To quantify the performance of trained agents, we report the expert-normalized scores on both training and test environments. To report training performance, we measure the average success rate of trained agents over 100 rollouts in training environments and divide it by the average success rate from the expert PPG agent used to collect demonstrations. For test performance, we train a separate expert PPG agent in test environments and compute expert-normalized scores in the same manner.

**Baseline and our method**  As a baseline, we consider *Instruct*RL [46], one of the state-of-the-art text-conditioned policies. *Instruct*RL utilizes a transformer-based policy and pre-trained M3AE [22]

---

[2]`https://github.com/openai/CLIP`

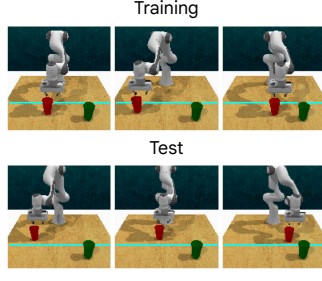

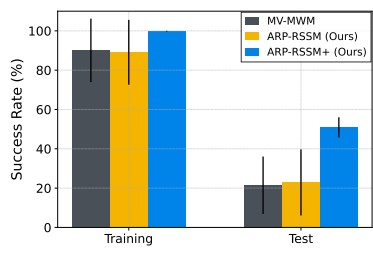

(a) Pick Up Cup

(b) Training/test performance

Figure 4: (a) Image observation of training and test environments for Pick Up Cup task in RLBench benchmarks [34]. (b) Success rates on both training and test environments. The result represents the mean and standard deviation over four different seeds. ARP-RSSM denotes the model that uses frozen CLIP representations for computing the multimodal reward, and ARP-RSSM+ denotes the model that incorporates fine-tuning scheme in Section 3.3.

representations for encoding visual observations and text instructions. For our methods, we use a return-conditioned policy based on Decision Transformer (DT) [8, 43] architecture, denoted as ARP-DT (see Appendix A for details). We consider two variations: the model that uses frozen CLIP representations (denoted as ARP-DT) and the model that uses fine-tuned CLIP representations (denoted as ARP-DT+) for computing the multimodal reward. We use the same M3AE model to encode visual observations and the same transformer architecture for policy training. The main difference is that our model uses sequence with multimodal return, while the baseline uses static text representations with the concatenation of visual representations.

**Comparison with language-conditioned agents** Figure 3 shows that our method significantly outperforms *Instruct*RL in all three tasks. In particular, ARP-DT outperforms *Instruct*RL in test environments while achieving similar training performance. This result implies that our method effectively guides the agent away from pursuing unintended goals through the adaptive multimodal reward signal, thereby mitigating goal misgeneralization. Moreover, we observe that ARP-DT+, which uses the multimodal reward from the fine-tuned CLIP model, achieves superior performance to ARP-DT. Considering that the only difference between ARP-DT and ARP-DT+ is using different multimodal rewards, this result shows that improving the quality of reward can lead to better generalization performance.

## 4.2 RLBench Experiments

**Environment** We also demonstrate the effectiveness of our framework on RLBench [34], which serves as a standard benchmark for visual-based robotic manipulations. Specifically, we focus on Pick Up Cup task, where the robot arm is instructed to grasp and lift the cup. We train agents using 100 expert demonstrations collected from environments where the position of the target cup changes above the cyan-colored line in each episode (see the upper figure of Figure 4a). Then, we evaluate the agents in a test environment, where the target cup is positioned below the cyan-colored line (see the lower figure of Figure 4a). The natural language instruction x used is "grasp the red cup and lift it off the surface with the robotic arm." For evaluation, we measure the average success rate over 500 episodes where the object position is varies in each episode.

**Setup** As a baseline, we consider MV-MWM [62], which initially trains a multi-view autoencoder by reconstructing patches from randomly masked viewpoints and subsequently learns a world model based on the autoencoder representations. We use the same procedure for training the multi-view autoencoders for our method and a baseline. The main difference is that while MV-MWM does not use any text instruction as an input, our method trains a policy conditioned on the multimodal return as well. In our experiments, we closely follow the experimental setup and implementation of the imitation learning experiments in MV-MWM. Specifically, we adopt a single-view control setup where the representation learning is conducted using images from both the front and wrist cameras, but world model learning is performed solely using the front camera. For our methods, we train the return-conditioned policy based on the recurrent state-space model (RSSM) [25], denoted

as ARP-RSSM (see Appendix A for more details). We consider two variants of this model: the model utilizing frozen CLIP representations (referred to as ARP-RSSM) and the model that employs fine-tuned CLIP representations (referred to as ARP-RSSM+). To compute multimodal rewards using both frozen and fine-tuned CLIP, we employ the same setup as in Procgen experiments. Additional details are in Appendix D.

**Results**   Figure 4b showcases the enhanced generalization performance of ARP-RSSM+ agents in test environments, increasing from 20.37% to 50.93%. This result implies that our method facilitates the agent in reaching target cups in unseen locations by employing adaptive rewards. Conversely, ARP-RSSM, which uses frozen CLIP representations, demonstrates similar performance to MV-MWM in both training and test environments, unlike the result in Section 4.1. We expect this is because achieving target goals for robotic manipulation tasks in RLBench requires more fine-grained controls than game-like environments.

## 4.3   Generalization to Unseen Instructions

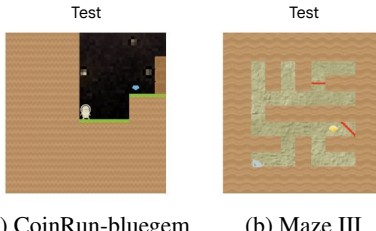

(a) CoinRun-bluegem     (b) Maze III

Figure 5: Test environments used for experiments in Section 4.3.

We also evaluate our method in test environments where the agent is now required to reach a different object with an unseen shape, color, and location by following unseen language instructions associated with this new object. First, we train agents in environments with the objective of collecting a yellow coin, which is always positioned in the far right corner, and learned agents are tested on

Table 1: Expert-normalized scores on CoinRun-bluegem test environments (see Figure 5a).

| Model | Test Performance |
|---|---|
| *Instruct*RL | 63.99% ± 3.07% |
| ARP-DT (Ours) | 77.05% ± 2.09% |
| ARP-DT+ (Ours) | 79.06% ± 6.69% |

unseen environments where the target object changes to a blue gem, and the target object's location is randomized. This new environment is referred to as CoinRun-bluegem (see Figure 5a), and we provide unseen instruction, "The goal is to collect the blue gem." to the agents. Table 1 shows that our method significantly outperforms the text-conditioned policy (*Instruct*RL) even in CoiunRun-bluegem. This result indicates that our multimodal reward can provide adaptive signals for reaching target objects even when the color and shape change.

In addition, we verify the effectiveness of our multimodal reward in distinguishing similar-looking distractors and guiding the agent to the correct goal. To this end, we train agents using demonstrations from Maze II environments, where the objective is to collect the yellow line. Trained agents are tested in an augmented version of Maze II test environments: we place a yellow gem, a red diagonal line, and a red

Table 2: Expert-normalized scores on Maze III test environments (see Figure 5b).

| Model | Test Performance |
|---|---|
| *Instruct*RL | 21.21% ± 1.52% |
| ARP-DT (Ours) | 33.33% ± 4.01% |
| ARP-DT+ (Ours) | 38.38% ± 3.15% |

straight line in the random position of the map (denoted as Maze III in Figure 5b), and instruct the trained agent to reach the red diagonal line ($\mathbf{x}$ ="Navigate a maze to collect the red diagonal line."). Table 2 shows that our method outperforms the baseline in Maze III, indicating that our multimodal reward can provide adaptive signals for achieving goals by distinguishing distractors.

## 4.4   Comparison with Goal-Conditioned Agents

We compare our method with goal-conditioned methods, assuming the availability of goal images in both training and test environments. First , it is essential to note that suggested baselines rely on additional information from the test environment because they assume the presence of a goal image during the test time. In contrast, our method relies solely on natural language instruction and

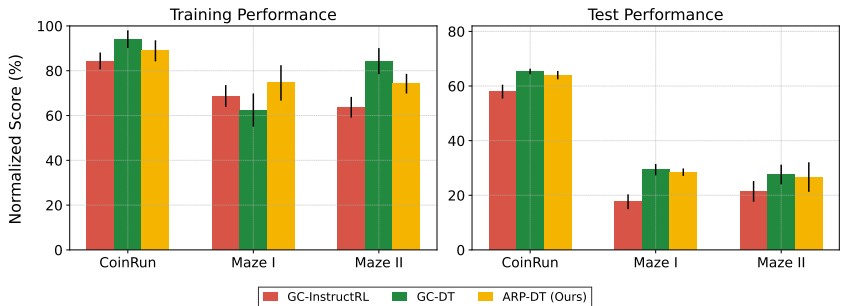

Figure 6: Expert-normalized scores on training/test environments. The result shows the mean and standard variation averaged over three runs. ARP-DT shows comparable or even better generalization ability compared to goal-conditioned baselines.

does not necessitate any extra information about the test environment. As baselines, we consider a goal-conditioned version of *Instruct*RL (denoted as GC-*Instruct*RL), which uses visual observations concatenated with a goal image at each timestep. We also consider a variant of our algorithm that uses the distance of CLIP visual representation to the goal image for reward (denoted as GC-DT).

Figure 6 illustrates the training and test performance of goal-conditioned baselines and ARP-DT. First, we observe that GC-DT outperforms GC-*Instruct*RL in all test environments. Note that utilizing goal image is the only distinction between GC-DT and GC-*Instruct*RL. This result suggests that our return-conditioned policy helps enhance generalization performance. Additionally, we find that ARP-DT demonstrates comparable results to GC-DT and even surpasses GC-*Instruct*RL in all three tasks. Importantly, it should be emphasized that while goal-conditioned baselines rely on the goal image of the test environment (which can be challenging to provide in real-world scenarios), ARP-DT solely relies on natural language instruction for the task. These findings highlight the potential of our method to be applicable in real-world scenarios where the agent cannot acquire information from the test environment.

## 4.5 Embedding Analysis

To support the effectiveness of our framework in generalization, we analyze whether our proposed method can induce meaningful abstractions in test environments. Our experimental design aims to address the key requirements for improved generalization in test environments: (i) the agent should consistently assign similar representations to similar behaviors even when the map configuration is changed, and (ii) the agent should effectively differentiate between goal-reaching behaviors and misleading behaviors. To this end, we measure the cycle-consistency of hidden representation from trained agents following [3, 42]. For two trajectories $\tau^1$ and $\tau^2$ with the same length $N$, we first choose $i \leq N$ and find its nearest neighbor $j = \arg\min_{j \leq N} ||h(o^1_{\leq i}, a^1_{<i}) - h(o^2_{\leq j}, a^2_{<j})||_2$, where $h(\cdot)$ denotes the output of the causal transformer of ARP-DT (refer to Appendix A for details). In a similar manner, we find the nearest neighbor of $j$, which is denoted as $k = \arg\min_{k \leq N} ||h(o^1_{\leq k}, a^1_{<k}) - h(o^2_{\leq j}, a^2_{<j})||_2$. We define $i$ as cycle-consistent if $|i - k| \leq 1$, can return to its original point. The presence of cycle-consistency entails a precise alignment of two trajectories within the hidden space.

In our experiments, we first collect the set of success/failure trajectories from $N$ different levels in CoinRun test environment, which is denoted as $\tau^n_{\text{succ}}$ or $\tau^n_{\text{fail}}$ where $n \in N$. Next, we extract hidden representations from trained agents at each timestep across all trajectories. We then measure cycle-consistency across these representations using three different pairs of trajectories (see Figure 7):

1. $(\tau^{n_1}_{\text{succ}}, \tau^{n_2}_{\text{succ}})$ ($\uparrow$): We compute the cycle-consistency between success trajectories from different levels. This indicates whether the trained agents behave in a similar manner in success cases, regardless of different visual contexts.

2. $(\tau^{n_1}_{\text{fail}}, \tau^{n_2}_{\text{fail}})$ ($\uparrow$): Similarly, we compute the cycle-consistency between failure trajectories from different levels.

3. $(\tau^{n_1}_{\text{succ}}, \tau^{n_1}_{\text{fail}})$ ($\downarrow$): We measure the cycle-consistency between success trajectory and failure trajectory from the same level. This evaluates whether the agent can act differently in success and failure cases.

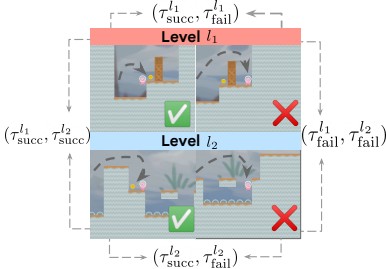

Figure 7: Visual observations of trajectories in CoinRun environments. We construct 3 different pairs to evaluate whether trained agents have well-aligned hidden representations.

Table 3: We investigate the cycle consistency of trained agents' hidden representations on different sets of trajectories in CoinRun environments. The results are presented as the mean and standard deviation averaged over three different seeds. Scores within one standard deviation from the highest average score are marked in bold. ($\uparrow$) and ($\downarrow$) denotes that higher/lower values are better, respectively.

| | *Instruct*RL | ARP-DT | ARP-DT+ |
|---|---|---|---|
| $(\tau_{\text{succ}}^{l_1}, \tau_{\text{succ}}^{l_2})$ ($\uparrow$) | 22.66% $\pm$ 4.26% | 24.75% $\pm$ 1.91% | **29.07%** $\pm$ **1.39%** |
| $(\tau_{\text{fail}}^{l_1}, \tau_{\text{fail}}^{l_2})$ ($\uparrow$) | 14.12% $\pm$ 2.18% | **24.95%** $\pm$ **0.46%** | **24.29%** $\pm$ **0.90%** |
| $(\tau_{\text{succ}}^{l_1}, \tau_{\text{fail}}^{l_1})$ ($\downarrow$) | 35.09% $\pm$ 2.05% | 30.18% $\pm$ 1.21% | **5.65%** $\pm$ **0.25%** |

Note that ($\uparrow$) / ($\downarrow$) implies that higher/lower value is better, respectively. For implementation, we first select ten different levels in CoinRun test environment with different coin locations. We then collect success and failure trajectories from each level and use the last ten timesteps of each trajectory for measuring cycle-consistency. Table 3 shows the percentage of timesteps that are cycle-consistent with other trajectories from different pairs. Similar to results described in Section 4.1, our proposed methods significantly improve cycle-consistency compared to *Instruct*RL in all cases. Moreover, ARP-DT+, which utilizes the multimodal reward from the fine-tuned CLIP model, outperforms ARP-DT with the frozen CLIP.

## 4.6 Ablation Studies

**Effect of pre-trained multimodal representations** To verify the effectiveness of pre-trained multimodal representations, we compare ARP-DT+ with agents using multimodal rewards obtained from a smaller-scale multimodal transformer, which was trained from scratch using VIP and IDM objectives, denoted as ARP-DT+ (scratch). Table 4 shows a significant decrease in performance for ARP-DT+ (scratch) when compared to ARP-DT+ across all environments, particularly in the training performance within Maze environments. These findings highlight the crucial role of pre-training in improving the efficacy of our multimodal rewards.

Table 4: Ablation study of using pre-trained CLIP representations.

| Env | Model | Train (%) | Test (%) |
|---|---|---|---|
| CoinRun | ARP-DT+ | 90.28% $\pm$ 1.59% | 72.36% $\pm$ 3.48% |
| | ARP-DT+ (scratch) | 77.08% $\pm$ 1.04% | 62.48% $\pm$ 5.32% |
| Maze I | ARP-DT+ | 75.47% $\pm$ 2.33% | 36.13% $\pm$ 0.78% |
| | ARP-DT+ (scratch) | 18.87% $\pm$ 5.87% | 32.52% $\pm$ 2.71% |
| Maze II | ARP-DT+ | 64.18% $\pm$ 3.62% | 40.95% $\pm$ 2.97% |
| | ARP-DT+ (scratch) | 22.51% $\pm$ 9.78% | 37.62% $\pm$ 8.73% |

**Effect of fine-tuning objectives** In Table 5, we examine the effect of fine-tuning objectives by reporting the performance of ARP-DT fine-tuned with or without the VIP loss $\mathcal{L}_{\text{VIP}}$ (Equation 3) and the IDM loss $\mathcal{L}_{\text{IDM}}$ (Equation 4). We find that the performance of ARP-DT improves with either $\mathcal{L}_{\text{VIP}}$ or $\mathcal{L}_{\text{IDM}}$, which shows the effectiveness of the proposed losses that encourage temporal smoothness and robustness to visual distractions. We also note that the performance with both objectives is the best, which implies that both losses synergistically contribute to improving the quality of the rewards.

Table 5: Ablation study of fine-tuning objectives: $\mathcal{L}_{\text{VIP}}$ and $\mathcal{L}_{\text{IDM}}$ in CoinRun.

| $\mathcal{L}_{\text{VIP}}$ | $\mathcal{L}_{\text{IDM}}$ | Train (%) | Test (%) |
|---|---|---|---|
| ✗ | ✗ | 89.58 % $\pm$ 2.08% | 63.32 % $\pm$ 2.01% |
| ✗ | ✓ | 89.24% $\pm$ 6.01% | 67.34 % $\pm$ 2.66% |
| ✓ | ✗ | 90.28% $\pm$ 2.17% | 70.35 % $\pm$ 1.01% |
| ✓ | ✓ | 90.28% $\pm$ 1.59% | 72.36 % $\pm$ 3.48% |

## 5 Conclusion

In this paper, we present Adaptive Return-conditioned Policy, a simple but effective IL framework for improving generalization capabilities. Our approach trains return-conditioned policy using the adaptive signal computed with pre-trained multimodal representations. Extensive experimental results demonstrate that our method can mitigate goal misgeneralization and execute unseen text instructions associated with new objects compared to text-conditioned baselines. We hope our framework could facilitate future research to further explore the potential of using multimodal rewards to guide IL agents in real-world applications.

## Acknowledgments and Disclosure of Funding

We want to thank Sihyun Yu, Junsu Kim, and anonymous reviewers for providing helpful feedback and suggestions for improving our paper. This research is supported by Institute of Information & Communications Technology Planning & Evaluation (IITP) grant funded by the Korea government (MSIT) (No.2022-0-00953, Self-directed AI Agents with Problem-solving Capability; No.2019-0-00075, Artificial Intelligence Graduate School Program (KAIST); No.2022-0-00184, Development and Study of AI Technologies to Inexpensively Conform to Evolving Policy on Ethics). This material is based upon work supported by the Google Cloud Research Credits program with the award (RDA2-TJCV-2DJT-HAHE).

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

# Supplementary Material

## Guide Your Agent with Adaptive Multimodal Rewards

## A  ARP Architecture Details

**Transformer-based policy (ARP-DT)**  To train agents following adaptive multimodal reward signals in Procgen experiments, we introduce a return-conditioned policy $\pi_\theta$ based on Decision Transformer architecture [8, 43]. Specifically, we train a decoder-only transformer to autoregressively model the following sequence:

$$\langle o_1, R_1, a_1, o_2, R_2, a_2, ..., o_T, R_T, a_T \rangle$$

Given the expert trajectory $\tau$, we first compute the target returns $\{R_i^*\}_{i=1}^T$ of expert demonstrations by computing the multimodal reward in Equation 1, and we tokenize the sequence using embedding layer for each modality. Our model $\pi$ parameterized by $\theta$ comprises the following components:

$$
\begin{aligned}
\text{Action decoder:} \quad & \pi_\theta(\hat{a}_t|o_{\leq t}, a_{<t}, R_{\leq t}) \\
\text{Return decoder:} \quad & \pi_\theta(\hat{R}_t|o_{\leq t}, a_{<t}, R_{<t})
\end{aligned}
\tag{5}
$$

All tokens are fed into the causal transformer, and it produces output embeddings. The action decoder receives the embedding and predicts action $\hat{a}_t$ and the return decoder takes the embedding for predicting multimodal return $\hat{R}_t$ to encourage the agent to be aware of the multimodal return. Following Lee et al. [43], we train the model to predict not only the next action but also the next multimodal return by minimizing the objective below:

$$\mathcal{L}_\pi(\theta) = \mathbb{E}_{\tau \sim \mathcal{D}} \left[ \sum_{t \leq T} (\pi_\theta(\hat{a}_t|o_{\leq t}, a_{<t}, R_{\leq t}), a_t^*) + \lambda \cdot \text{MSE}(\pi_\theta(\hat{R}_t|o_{\leq t}, a_{<t}, R_{<t}), R_t^*) \right], \tag{6}$$

where CE is the cross entropy loss, MSE is the mean squared error, and $\lambda$ is a hyperparameter that adjusts the scale of return prediction. We call ARP methods based on this architecture ARP-DT.

**RSSM-based policy (ARP-RSSM)**  To demonstrate the versatility of our framework across different model architectures, we introduce a return-conditioned policy based on world models [60, 62] for our RLBench experiments. Specifically, we implement the world model as a variant of the recurrent state-space model (RSSM; [25]) and condition it on multimodal return. The world model comprises the following components:

$$
\begin{aligned}
\text{Encoder:} \quad & s_t \sim f_\theta(s_t|s_{t-1}, a_{t-1}, o_t, R_t) \\
\text{Decoder:} \quad & \begin{cases} \hat{o}_t \sim p_\theta(\hat{o}_t \mid s_t) \\ \hat{R}_t \sim p_\theta(\hat{R}_t \mid s_t) \end{cases} \\
\text{Dynamics model:} \quad & \hat{s}_t \sim p_\theta(\hat{s}_t \mid s_{t-1}, a_{t-1}) \\
\text{Policy:} \quad & \hat{a}_t \sim p_\theta(\hat{a}_t|s_t)
\end{aligned}
\tag{7}
$$

The encoder extracts state $s_t$ from previous state $s_{t-1}$, previous action $a_{t-1}$, current observation $o_t$, and the target multimodal return $R_t = \sum_{i=t}^T R(o_i, \mathbf{x})$ with a recurrent architecture. The decoder reconstructs $o_t$ to provide a learning signal for model states and predicts $R_t$ to encourage the agent to be aware of the multimodal return. The policy predicts action $a_t$ using state $s_t$. All model parameters $\theta$ are optimized by minimizing the objective below:

$$
\begin{aligned}
\mathcal{L}(\theta) = \mathbb{E}_{\tau \sim \mathcal{D}} \Big[ & \sum_{t \leq T} - \ln p_\theta(z_t \mid s_t) - \ln p_\theta(R_t^* \mid s_t) - \ln p_\theta(a_t^*|s_t) \\
& + \beta \, \text{KL} \big[ q_\theta(s_t|s_{t-1}, a_{t-1}, z_t, R_t) \,\|\, p_\theta(\hat{s}_t|s_{t-1}, a_{t-1}) \big] \Big]
\end{aligned}
\tag{8}
$$

We refer to ARP methods based on this architecture as ARP-RSSM.

# B  Extended Related Work

**Training instruction-following agents**   Humans excel at understanding and utilizing language instructions to adapt to unfamiliar situations. Consequently, there has been significant interest in training policies that incorporate natural language in IL [48, 69, 36, 7] by learning policy conditioned on both current observation and the text instruction of the task. In parallel, recent studies have leveraged large language models (LLMs) for generating inner plans over pre-defined skills from natural language instructions for solving various robotic manipulation tasks [2, 45, 30, 27, 17]. Our method can be thought of as one of language-conditioned imitation learning, which leverages natural language instructions as a reward signal by utilizing the similarity between visual observations and natural language instructions in the pre-trained multimodal embedding space.

**Task specification with text instructions**   Leveraging text instructions for task guidance is a common practice among humans. Building upon this concept, prior approaches have harnessed text instructions to direct agents in various ways, including the shaping of reward functions [24, 75, 77, 40, 51], addressing misbehavior correction [65], and exploring human-AI coordination [75, 28]. The closest work to ours is Ma et al. [51], which extends the objective function of VIP [52] to include text instructions as goals for training visual-language aligned representations. The primary distinction is that it uses static text representation concatenated with visual representations for policy learning, akin to other baselines [46, 67]. In contrast, our approach employs multimodal reward defined by measuring the similarity between image observations and text instructions in the pre-trained multimodal embedding space.

**Utilizing CLIP [56] for supervision signals**   Recent work utilize CLIP scores [26] or CLIP-based perceptual loss [70] for improving image-text alignment in various domains including image generation [1, 12], image captioning [26, 9], and anomaly detection [37]. Similar to our approach, some work [13, 19] have also leveraged CLIP scores as supervision signals to address reward-scarce tasks with reinforcement learning. Fan et al. [19] propose a video-language model that is pre-trained using large-scale, real-world videos paired with their transcripts, and it utilizes the similarity between video-text representations as the reward for reinforcement learning. In our study, we focus on the adaptability of the multimodal reward, which empowers the agent to achieve desired goals in previously unseen test environments. Furthermore, we employ pre-trained multimodal representations without the need for resource-intensive pre-training, and we introduce a fine-tuning scheme for better reward quality that can be easily implemented with a small set of in-domain demonstrations.

# C  Procgen Experiment Details

This section describes the details for implementing Adaptive Return-conditioned Policy and provides our source code in the supplementary material.

**Environment details**   We utilize a publicly available implementation[3] to replicate the environments introduced by Di Langosco et al. [15]. We modify the simulator of the environments to render higher-resolution images to leverage pre-trained multimodal representations for both our method and baselines. In this particular setup, the observations obtained from the environment at each timestep $t$ comprise an RGB image with dimensions of $256 \times 256 \times 3$ and a natural language instruction delineating the desired goal. Throughout our experiments, we adhere to the *hard environment difficulty* as described in [10]. Maximum episode length for all tasks is 500. To gather expert demonstrations used for training data, we train PPG [11] agents on 500 training levels for 200M timesteps per task using hyperparameters provided in Cobbe et al. [11]. For evaluation purposes, we assess the test performance on 1,000 different levels, encompassing previously unseen themes and goals that differ from those employed in training.

**Architecture details**   Both *Instruct*RL (Liu et al., 2022) and ARP employ ViT-B/16 as the transformer-policy and pre-trained multimodal transformer encoder (M3AE; [22]) in all experiments, unless stated otherwise. For fine-tuning pre-trained multimodal encoders, we adopt the CLIP-Adapter [21, 80] to effectively fine-tune the pre-trained CLIP embeddings without overfitting.

---

[3]https://github.com/JacobPfau/procgenAISC

In detail, we attach extra linear layers to both the visual and text encoders and use the weighted sum of the output from the linear layers and the original pre-trained feature for computing visual/text representations. Throughout the training, we only apply gradients to the weight of these adapter layers and freeze both CLIP's visual and textual encoders. Furthermore, we utilize multi-scale features obtained by concatenating intermediate layer representations with the final output representation as the input for the adapter layers, drawing inspiration from Liu et al. [46] and Walmer et al. [71]. Finally, the multimodal reward is computed using the cosine similarity between the multi-scale features from the image and text encoders. See Appendix E for qualitative results of our multimodal rewards. Inspired by Gao et al. [21], we attach an additional 2-layer MLP to the end of a pre-trained multimodal transformer encoder and use the weighted sum of the output from MLP and the original pre-trained feature for obtaining image/text representations. In the training phase, we apply gradients only to the weight of these MLP layers. Through empirical evaluation, we observe that this architecture yields superior performance in both our method and the baseline.

**Training details** We resize images to $224 \times 224 \times 3$ before computing multimodal rewards, and we use $256 \times 256 \times 3$ RGB observations for training the return-conditioned policy. To stabilize training, we normalize multimodal returns following the method proposed by Chen et al. (2021), dividing them by 1000 in all experiments. We use the AdamW optimizer (Loshchilov et al., 2018) with a learning rate of $5 \times 10^{-4}$ and weight decay $5 \times 10^{-5}$. A cosine decay schedule is utilized to adjust the training learning rate. In CoinRun experiments, data augmentation techniques such as color jitter and random rotation are applied to the RGB images $o_t$ while maintaining alignment in the context. However, no augmentation is used to RGB images in Maze I/II experiments. For scaling the return prediction loss in training the return-conditioned policy, we set $\lambda = 0.01$ in CoinRun experiments and $\lambda = 0.001$ in Maze I/II experiments. During the fine-tuning of the pre-trained multimodal encoder, a 2-layer MLP is attached to the end of both CLIP image and text encoders. An extra 2-layer MLP is added as an action prediction layer for the IDM objective. The model is trained for 20 epochs, and the one with the lowest validation loss is used for generating multimodal rewards. To scale the IDM loss in fine-tuning CLIP, we employ $\beta = 1.5$ in CoinRun experiments and $\beta = 2.0$ in Maze I/II experiments. In evaluation, we choose the target multimodal return as 90% quantile of the multimodal return from the dataset $\mathcal{D}^*$ in all experiments.

**Computation** We use 24 CPU cores (Intel Xeon CPU @ 2.2GHz) and 2 GPUs (NVIDIA A100 40GB GPU) for training return-conditioned policy. The training of ARP for 50 epochs takes approximately 4 hours for CoinRun experiments with the largest dataset size. For fine-tuning CLIP, we use 24 CPU cores (Intel Xeon CPU @ 2.2GHz) and 1 GPU (NVIDIA A100 40GB GPU), which takes approximately 1.5 hours for Coinrun experiments.

**Hypeparameters** We report the hyperparameters used in our experiments in Table 6.

Table 6: Hyperparameters of ARP-DT. Unless specified, we use the same hyperparameters used in *Instruct*RL [46].

| Hyperparameter | Value |
|---|---|
| Policy batch size | 64 |
| Policy epochs | 50 |
| Policy context length | 4 |
| Policy learning rate | 0.0005 |
| Policy optimizer | AdamW [47] |
| Policy optimizer momentum | $\beta_1 = 0.9, \beta_2 = 0.999$ |
| Policy weight decay | 0.00005 |
| Policy learning rate decay | Linear warmup and cosine decay (see code for details) |
| Policy context length | 4 |
| Policy transformer size | 2 layers, 4 heads, 768 units |
| Fine-tuned CLIP batch size | 64 |
| Fine-tuned CLIP epochs | 20 |
| Fine-tuned CLIP learning rate | 0.0001 |
| Fine-tuned CLIP weight decay | 0.001 |
| Fine-tuned CLIP adapter layer size | 2 layers, 1024 units |
| Fine-tuned CLIP optimizer | AdamW [47] |
| Fine-tuned CLIP optimizer momentum | $\beta_1 = 0.9, \beta_2 = 0.999$ |

# D   RLBench Experiment Details

**Environment details**   Visual observations obtained from RLBench environment at each timestep $t$ comprise an RGB image with dimensions of 256 x 256 × 3 and a natural language instruction delineating the desired goal. In Pick Up Cup task, we adjusted the environment to generate the target cup with different regions in training and evaluation. Specifically, we generate the target cup from 80% of the area of the table in training environments. In evaluation, we generate the cup from the remaining 20% of the area of the table. We choose an action mode of RLBench that specifies the delta of joint positions for all experiments.

**Training details**   For training data, we collect 100 expert demonstrations using scripted policy provided by RLBench simulator. For each demonstration, we extract the keypoints [35] and train the agent to output the relative change in (x, y, z) position between keypoints. We train the agents to output the prediction of (x,y,z) position changes. We set the maximum episode length to 500. We closely follow the implementation details and use the same hyperparameters described in the imitation learning experiments in MV-MWM [62]. For all experiments, we utilize the open-sourced pre-trained CLIP model with ViT-B/16 architecture from huggingface transformers library[4] and we fine-tune CLIP based on that model. We train our method for 100K iterations and MV-MWM for 200K iterations until convergence. We apply data augmentation in training, including color jittering over RGB images. In evaluation, we choose the target multimodal return as 50% quantile of the multimodal return from training demonstrations in all experiments.

# E   Qualitative Results of Multimodal Rewards

In Figure 8, 9, 10, we present the curves of multimodal rewards for frozen/fine-tuned CLIP in the trajectories from training/held-out evaluation environments. We find that the multimodal reward exhibits an overall increasing trend as the agent approaches the goal in both frozen and fine-tuned CLIP, irrespective of the training and held-out evaluation environments. Furthermore, we observe that fine-tuned CLIP not only induces a reward that is temporally smoother in the intermediate stages compared to frozen CLIP (see Figure 1) but also demonstrates a steeper upward reward curve (see Figure 9, 10). These results support the claim that the quality of multimodal rewards from the fine-tuned CLIP outperforms those from the frozen CLIP (Section 4.1). Video examples of the trajectories are provided in the supplementary material.

---

[4]https://huggingface.co/openai/clip-vit-base-patch16

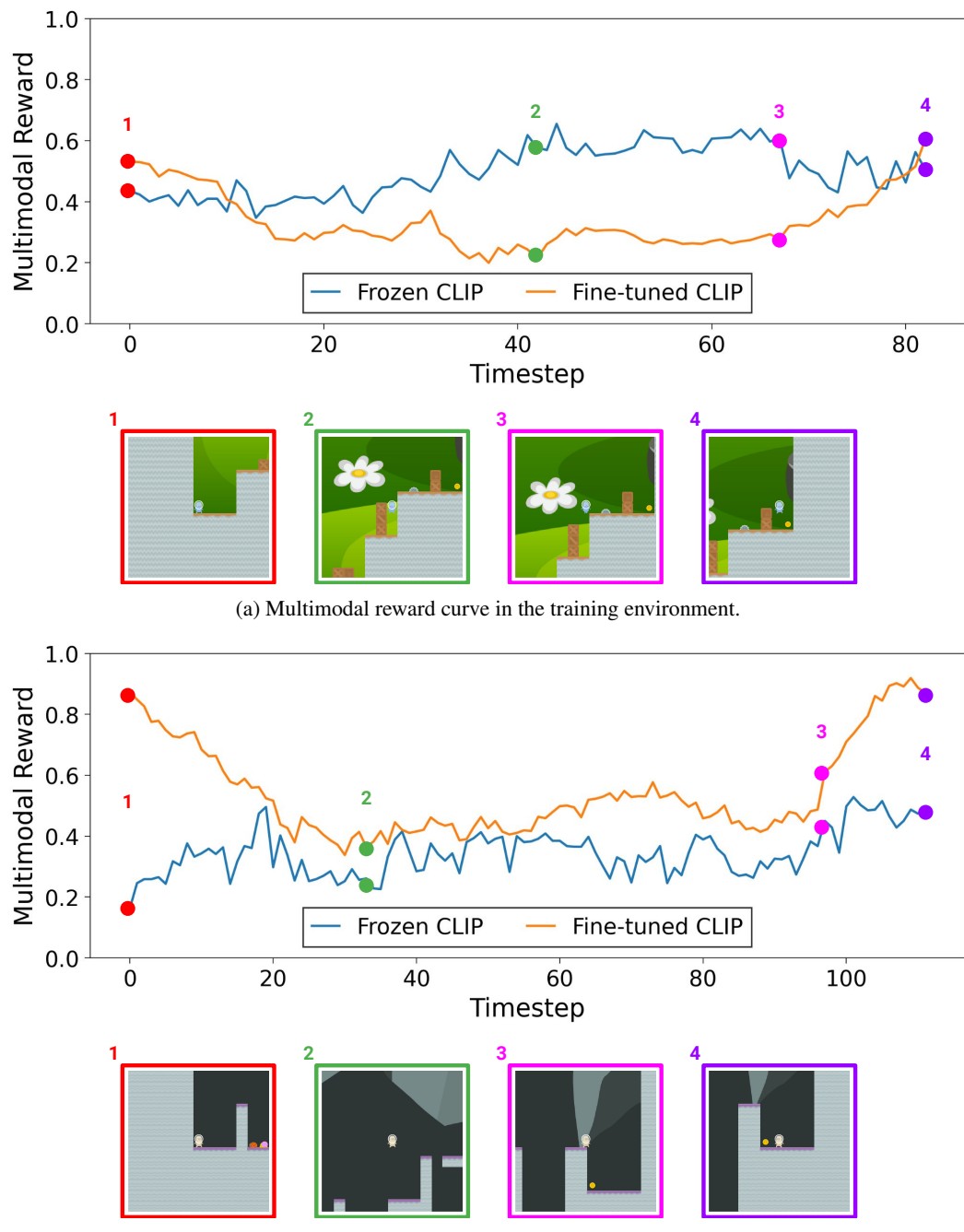

(a) Multimodal reward curve in the training environment.

(b) Multimodal reward curve in the held-out evaluation environment.

Figure 8: Qualitative results of multimodal rewards in CoinRun environments.

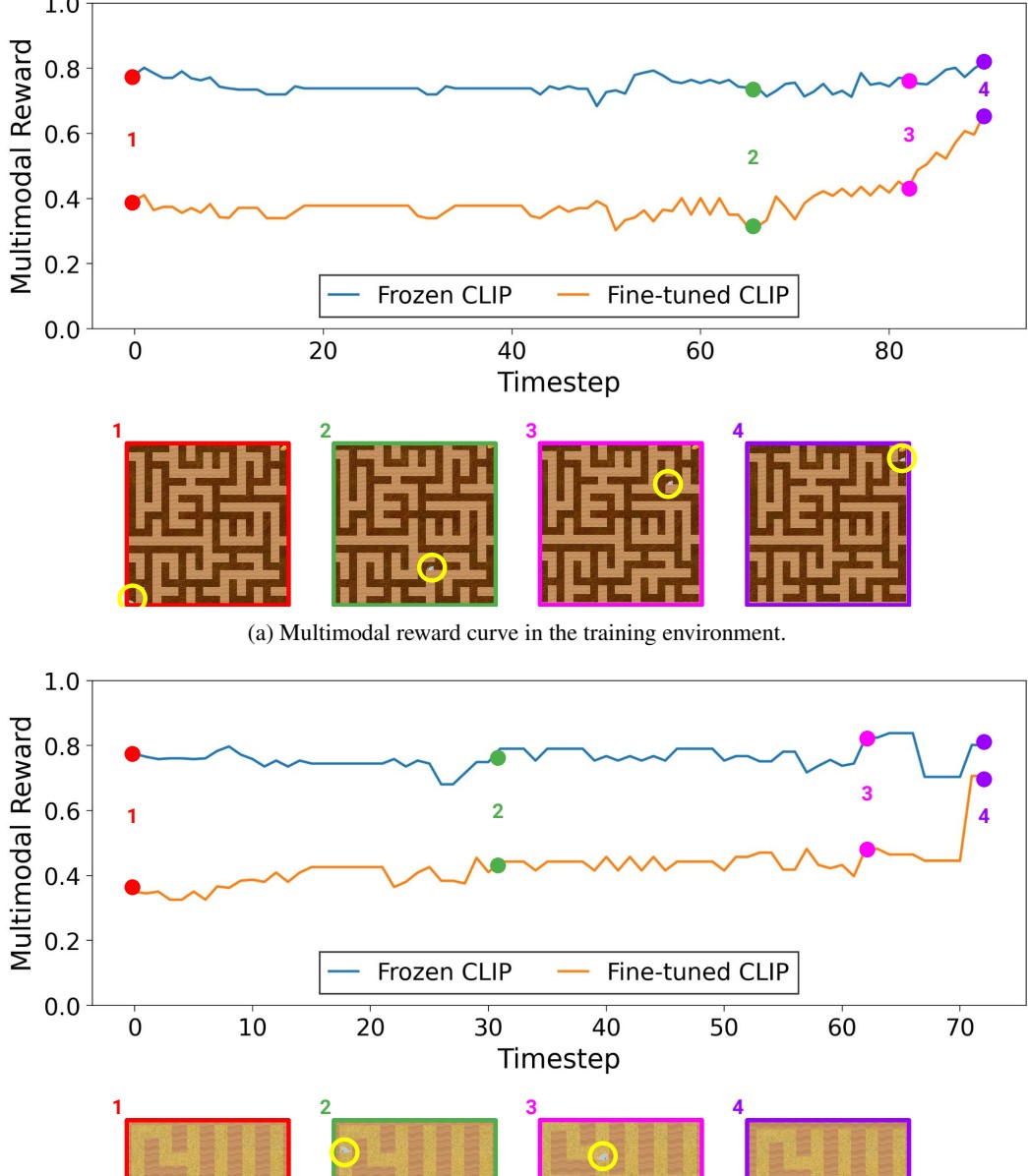

(a) Multimodal reward curve in the training environment.

(b) Multimodal reward curve in the held-out evaluation environment.

Figure 9: Qualitative results of multimodal rewards in Maze I environments.

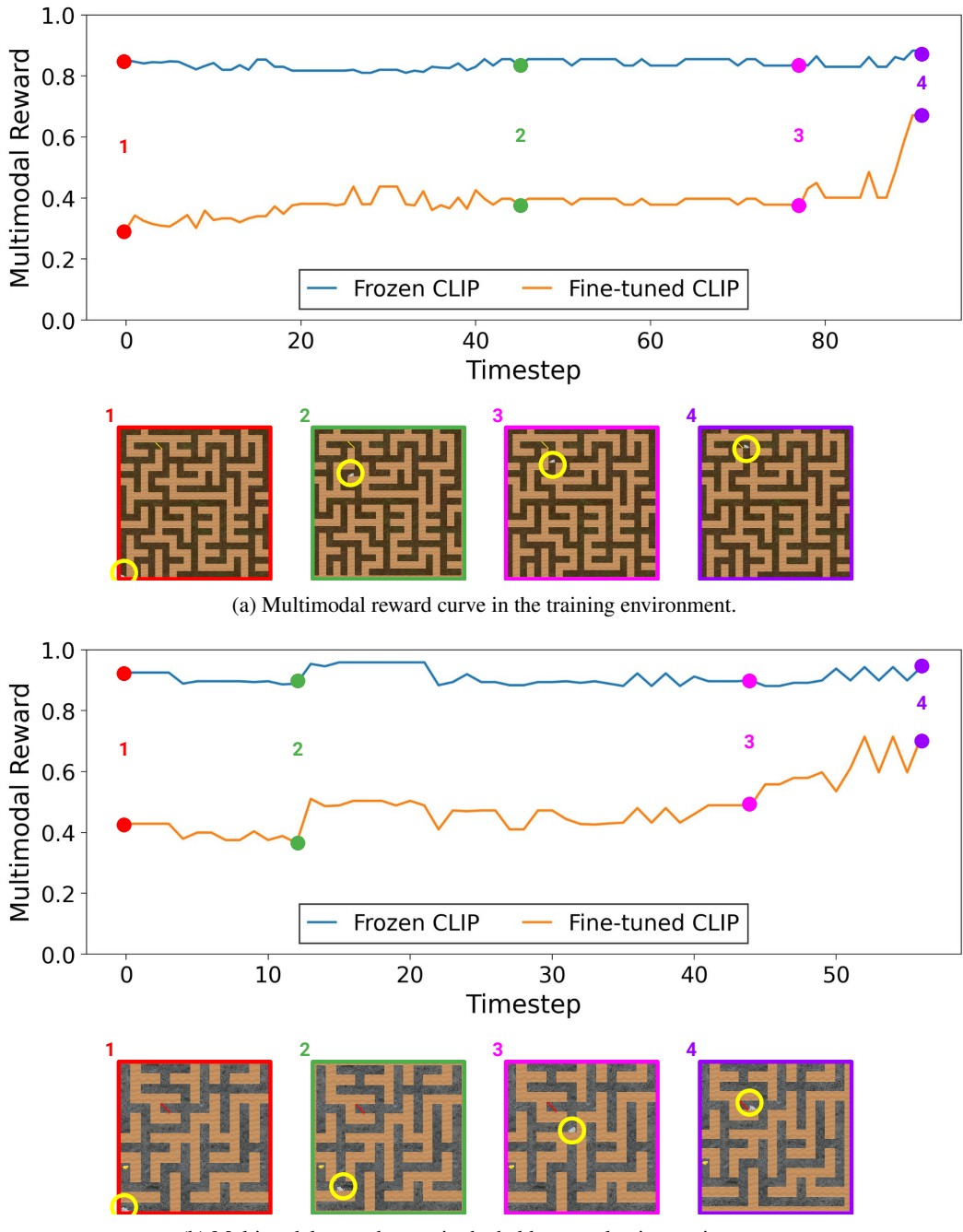

(a) Multimodal reward curve in the training environment.

(b) Multimodal reward curve in the held-out evaluation environment.

Figure 10: Qualitative results of multimodal rewards in Maze II environments.

# F    Additional Experiments

**Effect of return prediction**    We investigate the effect of including the return prediction loss $\text{MSE}(\hat{R}_t, \hat{R}_t^*)$ in Equation 6, which encourages the policy to be more aware of conditioned returns. In Table 7, we observe that the performance of ARP-DT becomes much more sensitive to the quality of multimodal rewards when trained with the return prediction loss. For instance, without the return prediction loss, the evaluation perfor-

Table 7: Ablation study of the return prediction loss $\text{MSE}(\hat{R}_t, \hat{R}_t^*)$ in CoinRun environments.

| $\text{MSE}(\hat{R}_t, \hat{R}_t^*)$ | $\mathcal{L}_{\text{FT}}$ | Train (%) | Test (%) |
|---|---|---|---|
| ✗ | ✗ | 90.28% ± 4.21% | 56.32% ± 3.55% |
|   | ✓ | 92.01% ± 3.18% | 56.28% ± 2.01% |
| ✓ | ✗ | 89.58% ± 2.08% | 63.32% ± 2.01% |
|   | ✓ | 90.28% ± 1.59% | 72.36% ± 3.48% |

mance becomes almost the same with or without the fine-tuning scheme, which suggests that the model is insensitive to the quality of rewards. On the other hand, with the prediction loss, the performance increases as the quality of reward improves. This implies that the model becomes aware of the returns and can thus follow the adaptive signal from the multimodal reward.

**Effect of scaling return prediction loss**    We investigate how the coefficient $\lambda$, which determines the weight of the return prediction loss in training return-conditioned policy, affects the performance of ARP-DT. To this end, we test various values of $\lambda$ in CoinRun environments. Table 8 shows the performance of ARP in training/held-out evaluation environments with different $\lambda$. We find that performance is not significantly different according to the value of $\lambda$ in the held-out evaluation environments. These results indicate that ARP is robust to the choice of hyperparameter $\lambda$.

Table 8: Ablation studies of the hyperparameter $\lambda$ in CoinRun environments.

| $\lambda$ | $\mathcal{L}_{\text{FT}}$ | Train (%) | Test (%) |
|---|---|---|---|
| 0.001 | ✗ | 89.93% ± 3.94% | 62.65% ± 10.12% |
|       | ✓ | 85.42% ± 1.80% | 71.69% ± 5.71% |
| 0.01 | ✗ | 89.58% ± 2.08% | 63.32% ± 2.01% |
|      | ✓ | 90.28% ± 1.59% | 72.36% ± 3.48% |
| 0.1 | ✗ | 87.15% ± 2.62% | 62.65% ± 10.31% |
|     | ✓ | 85.76% ± 3.18% | 73.37% ± 3.48% |
| 1.0 | ✗ | 87.15% ± 4.70% | 61.64% ± 6.38% |
|     | ✓ | 81.25% ± 1.04% | 73.37% ± 2.66% |

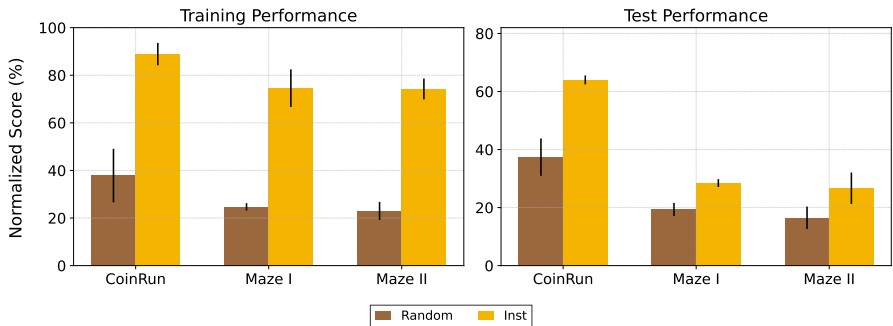

Figure 11: Expert-normalized scores on training/evaluation environments of ARP trained using multimodal rewards generated with (i) instructive text (*i.e.,* Inst) and (ii) random text (*i.e.,* Random). The result shows the mean and standard deviation averaged over three runs.

**Ablation studies on text instructions**    To investigate whether ARP-DT makes decisions based on the adaptive signal from the multimodal reward, we evaluate the quality of rewards generated with (i) instructive text (*i.e.,* Inst) and (ii) random text (*i.e.,* Random) in three different environments. Specifically, we use a natural language instruction for each environment, as described in Section 4 for Inst, and "NeurIPS 2023 will be held again at the New Orleans Ernest N. Morial Convention Center" for Random. We find that using random text instructions significantly declines performance in both training and evaluation environments. These results highlight the importance of using the instructive text and demonstrate that ARP-DT indeed depends on the adaptive signal from the multimodal reward for solving tasks at deployment time.

# G   Limitation and future work

One limitation of our work is that we currently rely on a single image-text pair to compute the multi-modal reward at every timestep $t$. Although our approach has shown effectiveness both quantitatively and qualitatively, there are tasks where rewards depend on the history of past observations (*i.e.,* non-Markovian) [4, 5, 39]. To address this limitation, it would be valuable to explore the extension of our method to incorporate video-text pairs for calculating multimodal rewards. Another aspect to consider is that the tasks we have examined so far are relatively simple, as they involve only a single condition for success. To tackle more complex problems, we are interested in investigating approaches that leverage large language models [29, 30, 2, 17] with our method.

# H   Potential negative impacts

We do not anticipate significant negative societal impacts in that our method is now limited to applying in simulated environments. However, if our method is applied in real-world scenarios, privacy concerns may arise, considering that behavior cloning agents used in such applications, like autonomous driving [63] or real-time control [7, 17], require large amounts of data, which often contain controversial information. Additionally, a behavior cloning policy presents a challenge as it imitates specified demonstrations, potentially including undesirable actions. If some evil actions are included in expert demonstrations (*e.g.,* behaviors that may be violent to the pedestrians are contained in the training data for mobile manipulation tasks), the policy could have significant negative impacts on users. To address this concern, future directions should focus on developing agents with safe adaptation in addition to performance enhancement efforts.

