# OpenReview forum: "Guide Your Agent with Adaptive Multimodal Rewards"
_NeurIPS.cc/2023/Conference — NeurIPS 2023 poster_

### Official Review · Reviewer_HCX5 · 2023-07-02

**Soundness:** 3 good
**Presentation:** 3 good
**Contribution:** 3 good
**Rating:** 6
**Confidence:** 4

**Summary:**

The paper proposes an interesting and novel idea of using large pretrained mulitmodal models to compute image-text alignment
score to use as a reward to train return-conditioned policies using a Decision Transformer (DT). They propose a novel
imitation learning framework called Multimodal Reward Decision Transformer (MRDT) which trains a return-conditioned policy
using adaptive reward signals from image-text multimodal rewards.

The authors also propose a fine-tuning scheme that uses VIP and IDM objective to fine-tune using CLIP-Adapter using data from in-doman demonstrations to improve performance.

The paper also shows MRDT successfully improves generalization to unseen levels and unseen goals on 3 environments from OpenAI procgen benchmark.

Authors also present an analysis on how MRDT helps mitigate goal misgeneralization and present a set of metrics to evaluate the quality of multimodal rewards.

**Strengths:**

1. Paper is well-written and easy to follow
2. The proposed approach is interesting and is shown to be effective on the OpenAI Procgen benchmark when generalizing to
unseen levels and unseen goals. It outperforms prior work InstructRL on all 3 environments by a decent margin.
The results also show that finetuning using proposed fine-tuning scheme further improves the results.
3. The analysis on quality of multimodal rewards is quite informative. It gives a clear insight on how the multimodal rewards differ
for different cases . The analysis shows finetuned multimodal rewards better capture distinction, distance (S) and distance(F).
4. The analysis in figure 8 shows the rewards generated with instructions aligned with the goal state gets higher reward compared
to a random instructions. This demonstrations a pretrained multimodal model like CLIP can give good reward signal to the policy.

**Weaknesses:**

1. The paper proposes an interesting idea but the experiment section is lacking breadth. I'd appreciate if authors consider evaluating their
method on multiple benchmarks which have multi-modal input. Similar to InstructRL paper authors can consider a subset of tasks from RLBench
for evaluation of the proposed method. The environments in ProcGen are quite simple and the generalization is only being tested for
 unseen instantiations of a single task (for 3 environments). Does MRDT work well if we want a single policy that can achive different goals?
One example is Object Navigation problem (but in simpler grid world environments), where the agent has to navigate to single instance of one of the n target object categories i.e. Find a chair/Find a sofa.
 It'd be nice if authors can add more experiments from different benchmarks to the paper.

2. It seems like for all 3 environments in ProcGen benchmark a better reward can be cosine between image embeddings of current state vs the expected goal state.
Have authors tried using this simple baseline? Does this lead to better or worse results than MRDT? The concern I have is that these environments are quite simple
and might not require multi-modal rewards.

3. Does using multimodal rewards from CLIP like model leads to goal misrepresentation problem? For example, let's consider a augmented version of Maze II environment.
If in addition to the yellow gem and diagonal line there was a straight red/yellow line and the task was ""Navigate a maze to collect the diagonal line"" or ""Navigate a maze to collect straight line"",
 how would all the baselines perform? Are these multi-modal rewards capable of clearly distinguishing these similar looking goals? or this leads to agents learning a average policy where it
 sometimes confuses the straight line with diagonal line because of misrepresented reward signal?

4. The qualitative results in appendix B are quite interesting and I appreciate authors added these results in the supplementary. In Figure 9, for the coinrun task the
Fine-tuned CLIP rewards are almost as high as the reward that agent achieves at the goal state. This seems concerning and hints towards multi-modal rewards not being able to distinguish
goal state from a random subset of states. This could lead to policy learning arbitary behaviors or not learning the task at all if trained/finetuned using RL instead of imitation learning.
Do authors have any insights on why the finetuned reward model show unexpected behavior? And how can this be mitigated so that we can successfully use these rewards for online training/finetuning

**Questions:**

1. The proposed method is interesting and a novel but intuitive idea to leverage pre-trained multi-modal models as rewards for learning a policy.
The idea is novel and seems promising, the main concern I have is on the breadth of the experiment section. It'd be nice if authors can show results on slightly
 more complex environments to support the effectiveness of MRDT. I'd be open to updating my rating if authors can demonstrate the effectiveness of MRDT on different benchmarks/tasks.
2. As outlined in the weakness section, it would be great if authors can discuss more about why the reward spikes initially for finetuned CLIP model as shown in the analysis in appendix B.
3. I'd appreciate if the authors can also discuss the goal misrepresentation problem I describe in point 3 of the weakness section.

**Limitations:**

1. The experiments section seems lacking at the moment and could benefit from evaluation on different benchmarks
2. Multi-modal rewards post finetuning exhibit unexpected behavior which could prove to be a big problem when finetuning policies
using these rewards. These rewards can sometimes misrepresent the value of the state agent is in.
3. It is not clear how well the approach would work in more complex tasks with partial observability and non-markovian states.

---

> ### Author Rebuttal · Authors · 2023-08-09
>
> Dear Reviewer HCX5,
>
> We sincerely appreciate your valuable comments. We found them extremely helpful in improving our draft. We address each comment in detail, one by one below.
>
> ---
>
> **[W1, L1] Limited experiments.**
>
> In response to your concern, we have investigated the efficacy of our method within RLBench in global response. As shown in Figure 1b in the attached pdf, we observe that our method can guide the agent to achieve unseen goals also in robotic manipulation tasks requiring fine-grained controls by incorporating our fine-tuning schemes. e.g., MR-RSSM+ achieves a performance of 50.93%, whereas MV-MWM achieves only 20.37%. Please refer to [GR1] in the global response for more details.
>
> In terms of object navigation problems you suggested, it would be an interesting future direction to investigate the effectiveness of our framework in multi-task setups.
>
> ---
>
> **[W2] Comparison with goal-conditioned baselines.**
>
> Thank you for your intriguing question. First of all, it is important to note that the suggested baseline relies on additional information from the test environment because it assumes availability of a goal image (or state) during the test time. On the other hand, our method solely relies on natural language instruction and does not require any extra access to the test environment. Despite not requiring any additional information, we observe that our method (MRDT) achieves comparable generalization performance to goal-conditioned baselines (GC-DT). Please refer to [GR2] in the global response.
>
> ---
>
> **[W3, Q3, L2] Goal misrepresentation problem that can be caused by using multimodal rewards.**
>
> Thank you for pointing this out. To address your point, we have evaluated agents trained using demonstrations from Maze II environments, where the objective is to collect the yellow line (see Figure 5a in the attached pdf), in an augmented version of Maze II test environments. Specifically, we place a yellow gem, a red diagonal line, and a red straight line in the random position of the map, and instruct the trained agent to reach the red diagonal line. We denote this environment as Maze III. The goal of this evaluation is to verify that our multimodal reward clearly distinguishes similar looking distractors and guides the agent to the correct goal. Please refer to Figure 5c in the attached pdf for image observation of Maze III.
>
> As depicted in the table presented below, we observe that our method outperforms the baseline also in Maze III. This result demonstrates that our multimodal reward can provide adaptive signals for achieving the goal by distinguishing distractors using natural language instructions.
>
> | Normalized Score | Maze II (Test)      | Maze III (Test)       |
> |------------------|-----------------|-----------------|
> | InstructRL       | 21.90% +- 3.30% | 21.21% +- 1.52% |
> | MRDT (Ours)      | 26.67% +- 5.41% | 33.33% +- 4.01% |
> | MRDT+ (Ours)     | 40.95% +- 2.97% | 38.38% +- 3.15% |
>
> ---
>
> **[W4, Q2, L2] Unexpected behavior of the fine-tuned reward model.**
>
> The unexpected behavior observed in multimodal rewards in fine-tuned CLIP in Figure 9 of the submitted draft could be attributed to the existence of round obstacles which have a similar shape to coin. (Please refer to the first frame of Figure 9b in the submitted draft.) However, as shown in Figure 9, 10, 11 of the submitted draft, our multimodal reward consistently exhibits an identifiable increasing pattern as the agent progresses toward the goal. Moreover, the issue of learning irrelevant behaviors might be mitigated through the utilization of our return-conditioned policy, as it is built upon a sequential architecture (DT [1], RSSM [2]). This architecture allows for the interpretation of the same reward value within the context of the agent's past trajectory. This contextual understanding can potentially alleviate the unexpected behavior and contribute to more effective utilization of the rewards for online training and finetuning purposes.
>
> ---
>
> **[L3] How well the approach would work in more complex tasks (partial observability, non-Markovian states)?**
>
> As shown in Figure 1 in the attached pdf, we observed that our method can guide the agent to achieve unseen goals in RLBench environment by incorporating our fine-tuning schemes. Considering that robotic manipulation tasks require more fine-grained control for gripping the object compared to Procgen environments, it remarks that our framework can be applicable in more complex tasks. However, our method does possess a limitation in addressing non-Markovian states, as the multimodal reward computation relies on image-text pair representations, potentially missing spatio-temporal relationships. An intriguing avenue for exploration is the extension of our approach to incorporate video-text pairs for multimodal reward computation. This limitation is discussed further in Appendix F of our draft.
>
> ---
>
> **References**\
> [1] Decision Transformer: Reinforcement Learning via Sequence Modeling, NeurIPS 2021.\
> [2] Learning Latent Dynamics for Planning from Pixels, ICML 2019.

---

> > ### Comment · Reviewer_HCX5 · 2023-08-17
> > **Response to Rebuttal**
> >
> > Thank for the responses.
> >
> > W1, L1: I appreciate the additional experiments. The results on RLBench are promising, please include these results in the final manuscript.
> > W2: Can you clarify what the input to the MRDT baseline for these experiments were? Was it a simple language instruction like "find red diagonal line" or "find yellow cheese"? And these goals used during test time are all already seen during training? If yes, would it be possible to just evaluate GC-DT and MRDT on ProcGen environments using an unseen goal? For example: evaluate MRDT trained to navigate to yellow cheese to go to a red diagonal line and similarly evaluate GC-DT with an image goal of red diagonal line in test environment? Such an evaluation will be more robust compared to evaluating MRDT with already seen goals vs GC-DT with unseen goals i.e. image goals from the test environment. I understand the short time left in the discussion period, in case authors can't provide results I'd appreciate some discussion on the evaluation setup of GC-DT vs MRDT.
> > W3, Q3, L2: I appreciate the experiments. It'd be nice to include these experiments in the appendix of manuscript as well.
> >
> > I think most of my concerns have been addressed apart from a discussion on evaluation setup on W2. I am updating my score to reflect this. I'd appreciate if authors can respond to my questions on evaluation setup  on W2.

---

> > > ### Author Response · Authors · 2023-08-17
> > > **Thank you for your response**
> > >
> > > Dear reviewer HCX5,
> > >
> > > Thank you for your reply! We’re pleased to hear that our response addressed your questions. We address your question about the setup for comparison with goal-conditioned baselines in below.
> > >
> > > **[W2] Setup of comparison with goal-conditioned baselines.**\
> > > For comparison with goal-conditioned baselines in [GR2], We would like to emphasize that we follow the same setup and use the same language instruction of Procgen experiment, which is mentioned in Section 4.1 of the submitted draft. To assess agents in test environments with unseen goals, we assess both MRDT and GC-DT in Maze II environments. The training dataset consists of expert demonstrations in which the agent approaches a yellow diagonal line, and we then evaluate in a modified environment with two unseen objects: a yellow gem and a red diagonal line, where the goal of the agent is to reach the red diagonal line (see Figure 2c of the submitted draft). We observe that MRDT shows comparable performance to GC-DT even when the goal for solving tasks is unseen in the training (see Figure 2 of the attached pdf).
> > >
> > > If you have any other questions or suggestions, please do not hesitate to let us know.
> > >
> > > Thank you very much,\
> > > Authors

---

> > > > ### Comment · Reviewer_HCX5 · 2023-08-17
> > > >
> > > > Thank you for answering my question, all of my concerns have been addressed by the discussion.

---

### Official Review · Reviewer_nhjK · 2023-07-07

**Soundness:** 2 fair
**Presentation:** 3 good
**Contribution:** 2 fair
**Rating:** 6
**Confidence:** 3

**Summary:**

The authors propose a formulation of the reward-conditioned decision transformer but do so using a learned reward function estimated by a pretrained CLIP style model which maps expert demonstrations and textual descriptions of these demonstrations into the same domain. They evaluate their method on the coinrun and Maze navigation domains.


**Strengths:**

1. The method is principled and the intuitive: The authors describe how to finetune existing language and vision models and use alignment between the observation and the text a reward for a decision transformer.
2. The method figures are clear and easy to understand (except Fig1 see weaknesses).


**Weaknesses:**

1. Example poorly motivated: How is goal misgeneralization any different from policies overfitting to training environments? Figure 1 needs improvement since the whole paper is motivated by it and currently it is impossible to tell what is happening in the 2 panes.
2. Comparison with baselines: The idea of using CLIP for reward generation is not entirely new and Zest https://arxiv.org/pdf/2204.11134.pdf is a very close related work published about a year ago. Comparing to this paper is a natural baseline.
3. Generalization across language commands : Does the method enable generalization using language commands? The only experiments are demonstrated are domain generalization (unclear how OOD the test tasks are in comparison to the train).
4. Unclear how the method prevents goal misgeneralization: While the introduction motivates the work from this problem, the experimental section does not lend any credence to the fact that this method would prevent such a problem. Experiment


**Questions:**

Please help me understand how to distinguish goal misgeneralization is different from overfitting to training tasks. Further, it's not clear to me how adding language prevents overfitting. Consider an extreme case, where the dataset of demonstrations only contains trajectories moving towards the coin placed in the exact same location. How does adding language prevent overfitting here? The language label here will still be the same across trajectories. It's unclear to me why language would prevent overfitting.

**Limitations:**

The paper evaluation is a bit tenuous and I would encourage expanding the scope of the experiments. Currently, a lot of the experiments are geared towards evaluating the reward. Since a pretrained model, i.e., CLIP is being used to generate the rewards, it would be useful for readers to know how much of a domain alignment is needed for this work and to what extent the method breaks when trained on vastly OOD environments.

---

> ### Author Rebuttal · Authors · 2023-08-09
>
> Dear Reviewer nhjK,
>
> We sincerely appreciate your valuable comments. We found them extremely helpful in improving our draft. We address each comment in detail, one by one below.
>
> ---
>
> **[W1] Clarification of Figure 1 in the submitted draft.**
>
> Figure 1 in the submitted draft presents an illustrative example of goal misgeneralization. In this scenario, an agent trained to collect a coin consistently positioned in the far right corner (left) might learn a behavior that directs it toward the far right corner of the level, even if the actual task reward comes from collecting the coin. As a result, the agent achieves a low task return when the location of the coin is randomized by heading to the far right of the level (right).
>
> ---
>
> **[W2] Comparison with other baseline (ZeST).**
>
> Following your suggestion, we have devised a variant of our method in which the reward is computed based on the goal image, opposed to utilizing natural language instructions, denoted as GC-DT. It is considered as an improved version of ZeST + DT [1] in that it utilizes M3AE representations and additionally optimizes return-prediction objective. Despite not requiring any information from the test environment, MRDT shows comparable generalization performance to goal-conditioned baselines. For more details, please refer to [GR2] in the global response.
>
> ---
>
> **[W3-1] Generalization across language commands.**
>
> We would like to remark that we investigated how language command affects the generalization performance in Section 4.3 of the submitted draft. Specifically, our observations indicate that the utilization of instructive text results in significantly improved generalization performance compared to employing random text.
>
> Nonetheless, following your suggestion, we have evaluated agents trained using demonstrations from Maze I environments, where the goal is to reach the yellow cheese, in unseen Maze II environments, where the target object changes to a yellow line. Note that the only distinction is that the agent now needs to reach a different object and receive different natural language instructions. As depicted in the table presented below, we observe that our method outperforms the baseline, demonstrating that our method achieves enhanced generalization through the understanding of language commands.
>
> | Normalized Score | Maze II (Test)  |
> |------------------|-----------------|
> | InstructRL       | 43.36% +- 1.36% |
> | MRDT (Ours)      | 47.88% +- 6.11% |
> | MRDT+ (Ours)     | 49.23% +- 2.82% |
>
> ---
>
> **[W3-2] How do out-of-distribution (OOD) test environments compare to the training environments?**
>
> Regarding the test environment, our evaluation encompasses not only unseen configuration of the map but also previously unseen goals (i.e., different goal position or goal object) that differ from those used in training.
>
> ---
>
> **[W4, Q2] How does our method prevent goal misgeneralization?**
>
> As you pointed out, prior approaches [2,3] have a drawback that the agent receives the same signal at every timestep from text instruction, manifested as single hidden embedding from pre-trained representation space. In contrast, our method computes the multimodal reward between current image observation and text instruction at every timestep. This reward dynamically adjusts based on visual observations, enabling it to guide the agent to achieve the goal even in unseen test environments (see Figure 9, 10, 11 in the submitted draft). Subsequently, we train a return-conditioned policy using demonstrations with reward labels, which makes decisions based on the adaptive multimodal reward signal. As a result, our approach facilitates an enhanced ability to generalize across different goals and effectively prevents goal misgeneralization within test environments. Please refer to Figure 4 in the attached pdf for visual descriptions.
>
> ---
>
> **[Q1] Distinguishing goal misgeneralization with overfitting to training tasks.**
>
> Goal misgeneralization [4,5] is a special case of overfitting to training tasks. It arises when an agent is overfitted to a misleading goal that isn't connected to the actual source of the true task reward, resulting in the agent achieving a low task return in the test environment.
>
> ---
>
> **[L1-1] Degree of domain alignment required for pre-trained models**
>
> Tasks demanding delicate control, such as manipulation, would require a more precise multimodal reward achieved through fine-tuning of pre-trained multimodal representations. For instance, as shown in Figure 1 in the attached pdf, we observe that MR-RSSM, employing frozen CLIP representations, demonstrates similar performance to baseline in RLBench. On the other hand, MR-RSSM+, which integrates our fine-tuning scheme, achieved significantly enhanced performance in test environments. This outcome indicates that our fine-tuning scheme becomes more crucial in complex tasks like robotic manipulations.
>
> ---
>
> **[L1-2] Condition for breaking our method.**
>
> Given that our multimodal reward heavily relies on CLIP image-text pair representations, it might encounter challenges in handling language instructions that involve complex spatio-temporal relationships and relative positions of multiple objects., e.g., Pick up the leftmost of the three cups in RLBench environments. For solving this problem, extending our approach to incorporate video-text pairs for multimodal reward computation would be an interesting future direction.
>
> ---
>
> **References**\
> [1] Can Foundation Models Perform Zero-Shot Task Specification For Robot Manipulation?, L4DC 2022.\
> [2] Instruction-Following Agents with Multimodal Transformer, ArXiv 2023.\
> [3] Perceiver-Actor: A Multi-Task Transformer for Robotic Manipulation, CoRL 2022.
> [4] Goal Misgeneralization in Deep Reinforcement Learning, ICML 2022.\
> [5] Goal Misgeneralization: Why Correct Speciﬁcations Aren’t Enough For Correct Goals, ArXiv 2023.\

---

> > ### Comment · Reviewer_nhjK · 2023-08-10
> > **Response to Rebuttal**
> >
> > Thanks for your experiments and rebuttal. I still have some concerns which I list below.
> > W1. What you have described is overfitting. And if it is indeed overfitting, then are a slew of works that address overfitting in offline RL. Can you please clarify if you mean something else? If not, then there is no point in adding more jargon to a field.
> > W2. Thanks for the experiment.
> > W3. To clarify: does the training data only contain yellow cheese? If so, I am worried this experiment does not really tell us much about the role of language conditioning in generalization as the agent may only attend to the "yellow" portion of the language command. Could you perhaps run inference only on another object with a different color?

---

> > > ### Author Response · Authors · 2023-08-11
> > > **Thank you for your additional feedback**
> > >
> > > **[W1] Additional clarification of the goal misgeneralization.**
> > >
> > > Thank you for the thoughtful comment! Goal misgeneralization [1, 2] is a particular type of out-of-distribution robustness failure in RL, and can be considered a special case of overfitting. This problem was studied in Langosco et al [1], on various Procgen environments. A subsequent work [2] argued why even correct specifications can still cause the agent to pursue an undesired goal that leads to good performance in training environments, but bad performance in novel test situations.
> > >
> > > We agree with you that this is an example of overfitting, but the term is a little more specific and descriptive for the problem setting considered in this paper, i.e., the semantic and visual generalization of an instruction-conditioned agent in an embodied environment.
> > >
> > > We will revise our final draft to include this discussion (e.g. in Introduction section).
> > >
> > > **References**\
> > > [1] Goal Misgeneralization in Deep Reinforcement Learning, ICML 2022\
> > > [2] Goal Misgeneralization: Why Correct Specifications Aren't Enough For Correct Goals, ArXiv 2022.
> > >
> > > ---
> > >
> > > **[W3] Inference of the trained agents on different object + different color**
> > >
> > > Thank you for your suggestion. Following your suggestion, we have evaluated agents which ared trained in Maze I environments (where the goal is to reach the yellow cheese) on new unseen Maze environments (Maze-redline), where the target object changes to a **red line**. Notably, the agent is now required to reach a different object with an unseen color and it receives different natural language instruction (i.e., “navigate a maze to collect the red line”) corresponding to the new object.
> > >
> > > As depicted in the table below, **we observe that our method outperforms the baseline, even when the agent is confronted with target objects of unseen shape and color during training.**
> > >
> > > | Normalized Score | Maze-redline (Test) |
> > > |------------------|---------------------|
> > > | InstructRL       | 55.10% +- 5.13%     |
> > > | MRDT (Ours)      | 60.52% +- 1.56%     |
> > > | MRDT+ (Ours)     | 61.43% +- 4.14%     |
> > >
> > > In addition, to further validate the effectiveness of our method, we have conducted similar experiments on CoinRun. We train agents on environments with the objective of collecting a yellow coin, which is always positioned in the far right corner, and then we test them on unseen environment (CoinRun-BlueGem), where the target object changes to a **blue gem** and **the target object’s location is randomized.**
> > >
> > > As indicated in the table below, we observe that our method also significantly outperforms the baseline.
> > > | Normalized Score | CoinRun-BlueGem (Test) |
> > > |------------------|------------------------|
> > > | InstructRL       | 63.99% +- 3.07%        |
> > > | MRDT (Ours)      | 77.05% +- 2.09%        |
> > > | MRDT+ (Ours)     | 79.06% +- 6.69%        |
> > >
> > > These results show that our multimodal reward can provide adaptive signals for achieving the goal by understanding language command in more challenging conditions, and the trained agent makes decisions based on that without relying on spurious patterns.

---

> > > > ### Comment · Reviewer_nhjK · 2023-08-11
> > > > **Reply**
> > > >
> > > > Thanks for the additional experiments. I think my concerns have been addressed in general. Updating my score to reflect this.

---

> > > > > ### Author Response · Authors · 2023-08-14
> > > > > **Thank you for your response**
> > > > >
> > > > > Dear reviewer nhjK,
> > > > >
> > > > > Thank you for your reply! We're pleased to hear that our response effectively addressed your inquiries.\
> > > > > If you have any other questions or suggestions, please do not hesitate to let us know.
> > > > >
> > > > > Thank you very much,\
> > > > > Authors

---

### Official Review · Reviewer_9gB5 · 2023-07-08

**Soundness:** 2 fair
**Presentation:** 3 good
**Contribution:** 2 fair
**Rating:** 4
**Confidence:** 4

**Summary:**

This paper presents a framework called Multimodal Reward Decision Transformer (MRDT) that uses the visual-text alignment score from pre-trained vision-language models (after careful fine-tuning using the in-domain data) as the reward signals in visual-based reinforcement learning. Specifically, the authors propose to train a return-conditioned policy based on Decision Transformer (DT) with the improved reward signals learned by the tuned VLMs. The method is shown to generalize better than without the multimodal reward strategy in environments with unseen goals partly due to the knowledge captured in pre-trained models.

**Strengths:**

+ The proposed method leverage pre-trained VLMs in a simple yet effective way for reward learning.
+ This is an important direction to study foundational models for decision-making. Compared to directly using pre-trained visual embedding models, this strategy of reward learning is more promising so far, IMO.

**Weaknesses:**

- Limited experiments. The authors only perform their studies on three tasks in ProcGen, far from most published work on this subject.
- Due to the limited experiments, it might require higher technical novelties. However, the novelty is also limited since the proposed method is built on top of several existing methods (VIP, IDM, CLIP-Adapter, etc.)

I am open to raising my score given more experimental evidence.

**Questions:**

Need to also discuss some other existing relevant literature on this subject, e.g.,

[1] Reward Design with Language Models

[2] LIV: Language-Image Representations and Rewards for Robotic Control

**Limitations:**

The major limitation is the lack of adequate evaluations. The approach is an otherwise promising one.

---

> ### Author Rebuttal · Authors · 2023-08-09
>
> Dear Reviewer 9gB5,
>
> We sincerely appreciate your valuable comments. We found them extremely helpful in improving our draft. We address each comment in detail, one by one below.
>
> ---
>
> **[W1] Limited experiments.**
>
> In response to your concern, we have investigated the efficacy of our method within RLBench, which serves as a standard benchmark for visual-based robotic manipulations. We observe that our method can guide the agent to achieve unseen goals also in robotic manipulation tasks requiring fine-grained controls by incorporating our fine-tuning schemes. e.g., MR-RSSM+ achieves a performance of 50.93%, whereas MV-MWM achieves only 20.37%. Please refer to the [GR1] in the global response.
>
> ---
>
> **[W2] Discussion over existing relevant literature.**
>
> We would like to highlight that LIV [1] is a concurrent work to ours in that it was made public after the submission deadline of NeurIPS 2023. LIV shares similarities with our approach by extending the objective function of VIP [2] to include natural language instructions as goals for training visual-language aligned representations. The primary distinction lies in LIV's use of static text representation, concatenated with visual representations, for policy learning, akin to other baselines (e.g., InstructRL [3]). In contrast, our approach introduces a novel imitation learning framework, employing image-text alignment in the pre-trained multimodal embedding space as a reward to train a return-conditioned policy.
>
> "Reward Design" [4] is also relevant to our research as it specifies a proxy reward function using pre-trained representations. Notably, it differs from our work in that it utilizes Large Language Model (LLM) to define the proxy reward function through prompting instructions. In contrast, our method capitalizes on the similarity between text instructions and current visual observations within the pre-trained multimodal embedding space for the reward signal. Furthermore, while "Reward Design" converts LLM responses into binary rewards, our method employs adaptive reward signals that dynamically adjust at each timestep to guide agents in previously unseen environments.
>
> We will enhance the depth of our discussions in the final draft by incorporating relevant papers that explore the utilization of language instructions for shaping reward functions [5, 8, 9], addressing misbehavior correction [7], and investigating human-AI coordination [6, 10].
>
> ---
>
> **References**\
> [1] LIV: Language-Image Representations and Rewards for Robotic Control, ICML 2023.\
> [2] VIP: Towards Universal Visual Reward and Representation via Value-Implicit Pre-Training, ICLR 2023.\
> [3] Instruction-Following Agents with Multimodal Transformer, ArXiv 2023.\
> [4] Reward Design with Large Language Models, ICLR 2023.\
> [5] Language to Rewards for Robotic Skill Synthesis, ArXiv 2023.\
> [6] Inferring the Goals of Communicating Agents from Actions and Instructions, Theory-of-Mind Workshop @ ICML 2023.\
> [7] Correcting robot plans with natural language feedback, ArXiv 2022.\
> [8] Using natural language for reward shaping in reinforcement learning, IJCAI 2019.\
> [9] Inferring rewards from language in context, ACL 2022.\
> [10] Language instructed reinforcement learning for human-ai coordination, ICML 2023.

---

> > ### Comment · Reviewer_9gB5 · 2023-08-21
> > **Reviewer Response**
> >
> > Thanks for the rebuttal. [W2] addresses my concerns while the added experiment in [W1] partially does so. I still believe it requires more tasks to better reflect the efficacy of this approach. I would like to raise my score to the borderline, slightly leaning toward rejection, though.

---

> > > ### Author Response · Authors · 2023-08-21
> > > **Thank you for your response**
> > >
> > > Dear Reviewer 9gB5,
> > >
> > > Thank you for your response and we are happy to hear that we have addressed most of your concerns.
> > >
> > > Following your suggestion, we will include RLBench experiments and also include more experimental results in more diverse tasks in the final manuscript.
> > >
> > > If you have any further questions or suggestions, please do not hesitate to let us know.
> > >
> > > Thank you very much!\
> > > Authors

---

### Official Review · Reviewer_qrbz · 2023-07-24

**Soundness:** 3 good
**Presentation:** 4 excellent
**Contribution:** 3 good
**Rating:** 7
**Confidence:** 4

**Summary:**

This work tackles the goal misgeneralization problem in goal-conditioned RL agent setups. They propose to leverage pre-trained multimodal models, i.e., CLIP, to serve as a reward function that simultaneously estimate both text and image modalities for deciding an engineered reward for each time step. The authors train a transformer-based model to take a sequence of encoded state representation, estimated reward from CLIP, and historical actions, to predict the next reward and state representations for learning a return-conditioned policy. Additionally, to encourage smoother and more robust reward functions, they adopt value implicit pre-training and inverse dynamic model to incentivize better adapted rewards. The experimental results on OpenAI Procgen benchmarks demonstrate better generalization abilities over baselines without the proposed reward engineering scheme.

**Strengths:**

- Utilizing CLIP as an adaptive reward function is sound and easy to implement.
- The proposed idea is neat and should be transferable or up-scalable to more complex tasks and environments.
- The adoptions of both VIP and IDM are to the point for the proposed return-conditioned policy training.
- The reward evaluation protocols are well-motivated and justified.

**Weaknesses:**

- While the experimental setups are solid, the adopted environments (testbed tasks) are far from real-world imageries, and the adaptation may not utilize very well (and much) what CLIP has learned during the pre-training. More challenging environments are needed to justify whether the multimodal rewards that coming out from a pre-trained models are indeed useful. I.e., a valid baseline is to just train a small sizable multimodal transformer (with both VIP and IDM applied) and see if the pre-training alignment is indeed that important/useful.
- To really (and effectively) showcase the generalization ability, I suggest experiments that are conducted on generalizing to unseen (but perhaps visually similar) tasks as a testbed for the multimodal rewards. Using the testbeds this work adopt, for example, Maze I transferring to Maze II would be an interesting and insightful experimental setup.
- Literature reviews: using CLIP scores or CLIP-based perceptual loss [1] has been quite popular these days. The author should provide more in-depth discussion on related works along using pre-trained multimodal models as a strong supervision signal.

[1] Vinker, Yael, et al. "Clipasso: Semantically-aware object sketching." ACM Transactions on Graphics (TOG) 41.4 (2022): 1-11.

**Questions:**

- Why don’t you choose tasks and/or domains that are closer to CLIP’s training distribution (i.e., more real-world imageries), such as some egocentric embodied tasks or navigational agent?
- Do you think your method, or perhaps, CLIP-based rewards, are trustworthy enough to generalize well and replace the actual environmental rewards? I.e., can we assume we can simply adopt the engineered rewards and disregard the environmental success signals to train an agent and it still performs (maybe almost) on par with that using the environment reward. (I guess one will need to further train a termination signal from CLIP, too.)

**Limitations:**

- The authors did not explicitly point out the limitations of this work. I suggest the authors discuss more on this matter in both their proposed ideas and the execution of the experiments.

---

> ### Author Rebuttal · Authors · 2023-08-09
>
> Dear Reviewer qrbz,
>
> We sincerely appreciate your valuable comments. We found them extremely helpful in improving our draft. We address each comment in detail, one by one below.
>
> ---
>
> **[W1] Effect of pre-trained models.**
>
> To verify the effectiveness of pre-trained multimodal representations, we compare our method with agents using multimodal rewards obtained from a smaller-scale multimodal transformer, which was trained from scratch using VIP and IDM objectives, denoted as MRDT+ (scratch).  Figure 3 in global response shows a significant decrease in performance for MRDT+ (scratch) when compared to MRDT+ across all environments, particularly in the training performance within Maze environments. These findings highlight the crucial role of pre-training in improving the efficacy of our multimodal rewards.
>
> ---
>
> **[W2] Transferring tasks for proving generalization ability.**
>
> Following your suggestion, we have evaluated agents trained using demonstrations from Maze I environments, where the goal is to reach the yellow cheese, in unseen Maze II environments, where the target object changes to a yellow line. Note that the only difference is that the agent now needs to reach a different object and it receives different natural language instruction. As depicted in the table presented below, we observe that our method outperforms the baseline in this unseen transfer scenario. These results indicate that our approach effectively guides agents towards desired goals in unseen environments, even when there is a change in the accompanying text command.
>
> | Normalized Score | Maze II (Test)  |
> |------------------|-----------------|
> | InstructRL       | 43.36% +- 1.36% |
> | MRDT (Ours)      | 47.88% +- 6.11% |
> | MRDT+ (Ours)     | 49.23% +- 2.82% |
>
> ---
>
> **[W3] Literature review for recent work utilizing CLIP scores / CLIP-based perceptual loss.**
>
> Following your suggestion, we will revise our manuscript with a more comprehensive literature survey of utilizing CLIP scores.
>
> Recent work utilize CLIP scores [1] or CLIP-based perceptual loss [2] for improving image-text alignment in various domains including image generation [3,4], image captioning [1, 5], and anomaly detection [6]. Similar to our approach, some work [7,8] have also leveraged CLIP scores as supervision signals to address reward-scarce tasks with reinforcement learning. Fan et al. [8] proposes a video-language model that is pre-trained using large-scale, real-world videos paired with their transcripts, and it utilizes the similarity between video-text representations as the reward for reinforcement learning.
>
> In our study, we focus on the adaptability of the multimodal reward, which empowers the agent to achieve desired goals in previously unseen test environments. Furthermore, we employ pre-trained multimodal representations without the need for resource-intensive pre-training, and we introduce a fine-tuning scheme for better reward quality that can be easily implemented with a small set of in-domain demonstrations.
>
> ---
>
> **[Q1] Selecting tasks close to CLIP’s training distribution.**
>
> Following your suggestion, we evaluated our method in RLBench, which provides more realistic visual observations compared to Procgen. We observe that our method (MR-RSSM+, 50.93%) significantly outperforms the baseline (MV-MWM, 20.37%) in the unseen test environment. Please refer to the [GR1] in the global response.
>
> ---
>
> **[Q2] Comparison between CLIP-based rewards and actual task rewards.**
>
> In our study, we have demonstrated the enhanced generalization performance of our proposed method with only natural language instructions in various environments. However, our approach may exhibit limitations when confronted with non-Markovian states. Since the computation of current multimodal rewards relies on the representations of image-text pairs, it might not effectively capture complex spatio-temporal relationships. Extending our method to incorporate video-text pairs for computing multimodal reward would be an interesting future direction for solving more complex environments characterized by extended horizons and multiple objectives
> .
>
> ---
>
> **[L1] Clarification on the limitations of our work.**
>
> We discuss some limitations of our work in Appendix F. Apologies for the confusion. We will move it to the main text in the final version.
>
> A primary limitation of our method is its current reliance on a single image-text pair to compute the multimodal reward. This design could potentially be a limitation in dealing with complex tasks that demand an understanding of spatio-temporal relationships across past observations. An intriguing avenue for future research would involve extending our methodology to incorporate video-text pairs by utilizing pre-trained video-text multimodal representations.
>
> From an experimental perspective, our approach does exhibit some weaknesses. For instance, we introduce certain hyperparameters for fine-tuning pre-trained multimodal representations (e.g., $\gamma$ and $\beta$ as discussed in Section 3.3 of the submitted draft) that depend on the specific task. Additionally, the quality of the multimodal reward may prove sensitive to the choice of prompts.
>
>
> ---
>
> **References**\
> [1] CLIPScore: A Reference-free Evaluation Metric for Image Captioning, EMNLP 2021.\
> [2] CLIPasso: Semantically-Aware Object Sketching, ACM Transactions on Graphics (TOG), 41(4):1-1, 2022.\
> [3] StyleGAN-NADA: CLIP-Guided Domain Adaptation of Image Generators, ACM Transactions on Graphics (TOG), 41(4):1–13, 2022.\
> [4] Vqgan-clip: Open domain image generation and editing with natural language guidance, ECCV 2022.\
> [5] Fine-grained image captioning with CLIP reward, NAACL 2022.\
> [6] WinCLIP: Zero-/few-shot anomaly classification and segmentation, CVPR 2023.\
> [7] Can Foundation Models Perform Zero-Shot Task Specification For Robot Manipulation?, L4DC 2022.\
> [8] Minedojo: Building open-ended embodied agents with internet-scale knowledge, NeurIPS 2022.

---

> > ### Comment · Reviewer_qrbz · 2023-08-10
> >
> > Thank for the responses.
> >
> > I appreciate the Maze II transfer experiments. In my opinion, even if the models attend more on the color "yellow", without explicitly stating the object attributes it may still fail. But I do agree with another reviewer that the yellow color could be a spurious pattern the model capture, so simply changing the line color (to something farther from yellow) would strengthen up the point.
> >
> > For Q1, please do include them into your final manuscript. It's both surprising (and perhaps promising too) that the pretraining domain helps RLBench that much.

---

> > > ### Author Response · Authors · 2023-08-11
> > > **Thank you for your additional feedback**
> > >
> > > **[C1] Inference of the trained agents on different object + different color**
> > >
> > > Thank you for your suggestion. Following your suggestion, we have evaluated agents which ared trained in Maze I environments (where the goal is to reach the yellow cheese) on new unseen Maze environments (Maze-redline), where the target object changes to a **red line**. Notably, the agent is now required to reach a different object with an unseen color and it receives different natural language instruction (i.e., “navigate a maze to collect the red line”) corresponding to the new object.
> > >
> > > As depicted in the table below, **we observe that our method outperforms the baseline, even when the agent is confronted with target objects of unseen shape and color during training.**
> > >
> > > | Normalized Score | Maze-redline (Test) |
> > > |------------------|---------------------|
> > > | InstructRL       | 55.10% +- 5.13%     |
> > > | MRDT (Ours)      | 60.52% +- 1.56%     |
> > > | MRDT+ (Ours)     | 61.43% +- 4.14%     |
> > >
> > > In addition, to further validate the effectiveness of our method, we have conducted similar experiments on CoinRun. We train agents on environments with the objective of collecting a yellow coin, which is always positioned in the far right corner, and then we test them on unseen environment (CoinRun-BlueGem), where the target object changes to a **blue gem** and **the target object’s location is randomized.**
> > >
> > > As indicated in the table below, we observe that our method also significantly outperforms the baseline.
> > > | Normalized Score | CoinRun-BlueGem (Test) |
> > > |------------------|------------------------|
> > > | InstructRL       | 63.99% +- 3.07%        |
> > > | MRDT (Ours)      | 77.05% +- 2.09%        |
> > > | MRDT+ (Ours)     | 79.06% +- 6.69%        |
> > >
> > > These results show that our multimodal reward can provide adaptive signals for achieving the goal by understanding language command in more challenging conditions, and the trained agent makes decisions based on that without relying on spurious patterns.
> > >
> > > ---
> > >
> > > **[Q1]**
> > > For Q1, we will revise our final draft to include RLBench experiments in Experiment section.

---

### Author Rebuttal · Authors · 2023-08-09

Dear reviewers,

We sincerely appreciate your time and effort in reviewing our manuscript. As the reviewers highlighted, our work proposes a simple but effective framework (qrbz, HCX5, 9gB5) for imitation learning with a novel adaptive reward that leverages pre-trained multimodal representations (9gB5, HCX5), and demonstrates strong empirical performance (HCX5). Our paper also provides informative analyses (qrbz, HCX5) and clear presentations (HCX5, nhjK).

We appreciate the reviewers’ insightful comments on our paper. In the attached pdf, we have included the following additional results to address the reviewer’s comments:
- RLBench experiment (Figure 1)
- Comparison with goal-conditioned baselines (Figure 2)
- Ablation study of MRDT+: comparison with smaller multimodal transformer trained from scratch (Figure 3)
- Concept figure depicting the difference between MRDT and baseline (Figure 4)
- Image observation of Procgen Maze environments used in our additional experiments (Figure 5)

---

The main concerns raised by multiple reviewers were (1) limited experiments and (2) lack of comparison to prior work that utilizes goal images. For (1), we prepare additional results on RLBench [1]. For (2), we prepared additional experiments comparing our method with goal-conditioned baselines. We describe the details of each experiment below.

**[GR1] RLBench experiment**

To further verify the effectiveness of our method, we evaluate on RLBench [1], which serves as a standard benchmark for visual-based robotic manipulations. Specifically, we test on the Pick Up Cup task, where the robot arm is instructed to grasp and lift the cup. We train agents in environments where the position of the cup to lift changes in each episode, and then evaluate the agents in a test environment, where the target cup is placed in an unseen region (refer to Figure 1a in the attached pdf).

As a baseline, we consider MV-MWM [2]. We closely follow the experimental setup and implementation of the imitation learning experiments in MV-MWM. For training agents, we implement the return-conditioned policy using a variant of the recurrent state-space model (RSSM ; [4]), where the recurrent encoder is additionally conditioned on multimodal return. We refer to our model that utilizes frozen CLIP representations for computing the multimodal reward as MR-RSSM, and our model that incorporates the proposed fine-tuning scheme as MR-RSSM+.

Figure 1b in global response showcases the enhanced generalization performance of MR-RSSM+ agents in test environments, increasing from 20.37\% to 50.93\%. This result implies that our method facilitates the agent in reaching unseen goals by employing adaptive rewards on complex robotics tasks.

**[GR2] Comparison with goal-conditioned baselines**

We compare our method with goal-conditioned methods assuming the availability of goal images in both training and test environments. First of all, it is important to note that the suggested baseline relies on additional information from the test environment because it assumes the presence of a goal image (or state) during the test time. On the other hand, our method solely relies on natural language instruction and does not necessitate any extra access to the test environment.

For goal-conditioned baseline, we consider a variant of our algorithm that uses the distance of CLIP visual representation to the goal image as a reward, denoted as GC-DT. The training and test performance of the goal-conditioned baselines and MRDT are visually presented in Figure 2 in the attached pdf.

We observe that MRDT demonstrates comparable results to GC-DT in all three tasks. Importantly, it should be emphasized that while goal-conditioned baselines rely on the goal image of the test environment (which can be challenging to provide in real-world scenarios), MRDT solely relies on a natural language instruction for the task. These results show the potential of our method to be applicable in real-world scenarios where the agent cannot acquire information from the test environment.

---

We sincerely believe that our enhanced manuscript will deliver the benefits of our proposed method to the audience of NeurIPS much better.

Thank you very much!\
Authors.

**References**\
[1] RLBench: The Robot Learning Benchmark & Learning Environment, IEEE Robotics and Automation Letters, 5(2), 2020.\
[2] Multi-View Masked World Models for Visual Robotic Manipulation, ICML 2023.\
[3] Mastering Atari with Discrete World Models, ICLR 2021.\
[4] Learning Latent Dynamics for Planning from Pixels, ICML 2019.

---

### Decision · Program_Chairs · 2023-09-21

**Decision:**

Accept (poster)

**Comment:**

This paper presents a framework, Multimodal Reward Decision Transformer (MRDT), that leverages pre-trained vision-language models like CLIP for reward generation in goal-conditioned reinforcement learning (RL). It aims to address the issue of goal misgeneralization on the OpenAI Procgen benchmark. The reviewers had mixed opinions about the work, but after discussion, most concerns are addressed and reviewers lean towards the acceptance of this work.
Strengths highlighted include the novelty of using CLIP as an adaptive reward function, a potentially scalable approach, and methodological soundness. However, reviewers also emphasize various limitations, particularly around the breadth of experimental validation. There is a consistent call for tests on more complex or real-world tasks to assess the true capabilities and generalizability of MRDT. Despite these criticisms, the paper's strengths in pioneering the use of pre-trained multimodal models for RL and good experimental protocols merit its Accept status, although further validation is advised for future work.